# Sensitivity of Greenland ice sheet projections to spatial resolution in higher-order simulations: the AWI contribution to ISMIP6-Greenland using ISSM

Martin Rückamp[1], Heiko Goelzer[2,3,4], and Angelika Humbert[1,5]

[1]Alfred-Wegener-Institut Helmholtz-Zentrum für Polar- und Meeresforschung, Bremerhaven, Germany
[2]Utrecht University, Institute for Marine and Atmospheric Research (IMAU), Utrecht, the Netherlands
[3]Laboratoire de Glaciologie, Université Libre de Bruxelles, Brussels, Belgium
[4]NORCE Norwegian Research Centre, Bjerknes Centre for Climate Research, Bergen, Norway
[5]University of Bremen, Bremen, Germany

**Correspondence:** Martin Rückamp (martin.rueckamp@awi.de)

**Abstract.** Projections of the contribution of the Greenland ice sheet to future sea-level rise include uncertainties primarily due to the imposed climate forcing and the initial state of the ice sheet model. Several state-of-the-art ice flow models are currently being employed on various grid resolutions to estimate future mass changes in the framework of the Ice Sheet Model Intercomparison Project for CMIP6 (ISMIP6). Here we investigate the sensitivity to grid resolution on centennial sea-level contributions from the Greenland ice sheet and study the mechanism at play. We employ the finite-element higher-order Ice Sheet System Model (ISSM) and conduct experiments with four different horizontal resolutions, namely 4, 2, 1 and 0.75 km. We run the simulation based on the ISMIP6 core climate forcing from the MIROC5 global circulation model (GCM) under the high emission Representative Concentration Pathway (RCP) 8.5 scenario and consider both atmospheric and oceanic forcing in full and separate scenarios. Under the full scenarios, finer simulations unveil up to approximately 5% more sea-level rise compared to the coarser resolution. The sensitivity depends on the magnitude of outlet glacier retreat, which is implemented as a series of retreat masks following the ISMIP6 protocol. Without imposed retreat under atmosphere-only forcing, the resolution dependency exhibits an opposite behaviour with about approximately 5% more sea-level contribution in the coarser resolution. The sea-level contribution indicates a converging behaviour below 1 km horizontal resolution. A driving mechanism for differences is the ability to resolve the bedrock topography, which highly controls ice discharge to the ocean. Additionally, thinning and acceleration emerge to propagate further inland in high resolution for many glaciers. A major response mechanism is sliding, with an enhanced feedback on the effective normal pressure at higher resolution leading to a larger increase in sliding speeds under scenarios with outlet glacier retreat.

## 1 Introduction

Climate change is the major driver of global sea-level rise (SLR), which has been shown to accelerate (Nerem et al., 2018; Shepherd et al., 2019). The Greenland ice sheet (GrIS) has contributed about 20% to sea-level rise during the last decade (Rietbroek et al., 2016). Holding in total an ice mass of $\sim 7.42$ m sea-level equivalent (SLE) (Morlighem et al., 2017), its future

contribution poses a major societal challenge. Since 1992, the GrIS mass loss is controlled on average at 52% by surface mass balance (SMB), with the remainder of 48% being due to increased ice discharge of outlet glaciers into the surrounding ocean (Shepherd et al., 2019).

While the relative importance of outlet glacier discharge for total GrIS mass loss has decreased since 2001 (Enderlin et al., 2014; Mouginot et al., 2019) and is expected to decrease further in the future (e.g. Aschwanden et al., 2019), it remains an important aspect for projecting future sea-level contributions from the ice sheet on the centennial timescale. (Goelzer et al., 2013; Fürst et al., 2015). A (non-linear) dynamic response of the ice sheet is caused by changes in the atmospheric and oceanic forcing, that may trigger glacier acceleration and thinning of outlet glaciers. Moreover, processes such as SMB and

ice discharge are mutually competitive in removing mass from the ice sheet (Goelzer et al., 2013; Fürst et al., 2015). Beside this interplay, a simple extrapolation of observed GrIS mass loss trends over the next century is not justified, as high temporal variations in SMB and glacier acceleration are apparent (e.g. Moon et al., 2012). Therefore, reliable ice sheet models (ISMs) forced with future climate data must be driven for policy relevant sea-level projections on century time scales.

    The Ice Sheet Model Intercomparison Project (ISMIP6, Nowicki et al., 2016, 2020a) is an international community effort

striving to improve sea-level projections from the Greenland and Antarctic ice sheets. Based on previous efforts like SeaRISE (Bindschadler et al., 2013; Nowicki et al., 2013) and ice2sea (e.g., Gillet-Chaulet et al., 2012), ISMIP6 continues to fully explore the sea-level rise contribution and associated uncertainties. The effort is aligned with the Coupled Model Intercomparison Project Phase 6 (CMIP6, Eyring et al., 2016) to provide input for the upcoming assessment report of the Intergovernmental Panel on Climate Change (IPCC AR6). The general strategy is to use outputs from CMIP5 and CMIP6 climate models to

derive atmosphere and ocean fields for forcing ISMs. Goelzer et al. (2020a) and Nowicki et al. (2020b) analyse the future sea-level contribution from multi-model ensembles for ISMIP6-projection-Greenland. The major aim of Goelzer et al. (2020a) is to provide future sea-level change projections and related uncertainty in a consistent framework.

    Despite substantial progress in ice sheet modelling in the last decades and years, a challenging goal remains to narrow uncertainties and improve the reliability of future sea-level projections from the two big ice sheets. Up to date, it is recognized

that the largest uncertainty sources are related to the initialization of the ISM or stem from the external forcing (Goelzer et al., 2018, 2020a). Goelzer et al. (2018) compared the initialization techniques used by different ice sheet modelling groups. The schematic forward experiment was not designed to estimate realistic sea-level contribution, but it provides valuable insights how the initial state of an ISM affects the ice sheet response. Under a predefined SMB anomaly, mass losses reveal a large spread. Although the spread is attributed to the broad diversity in model approaches, initMIP-Greenland shows notable im-

provements (e.g., a reduced model drift) and more consistent results compared to earlier large scale intercomparison exercises.

    Interestingly, the estimated sea-level contributions show a dependence on grid resolution (Fig. 1). ISM versions with multiple grid resolutions demonstrate that coarser grid resolutions tend to produce a slightly larger mass loss. This effect is partly due to the methodological approach by considering a SMB anomaly that is based on the present-day observed SMB. That means ISMs with initial areas larger than observed are subject to more and stronger melting and sharper transitions in SMB. Therefore,

coarse resolution models not rendering the present-day ice margin perfectly will likely overestimate ablation.

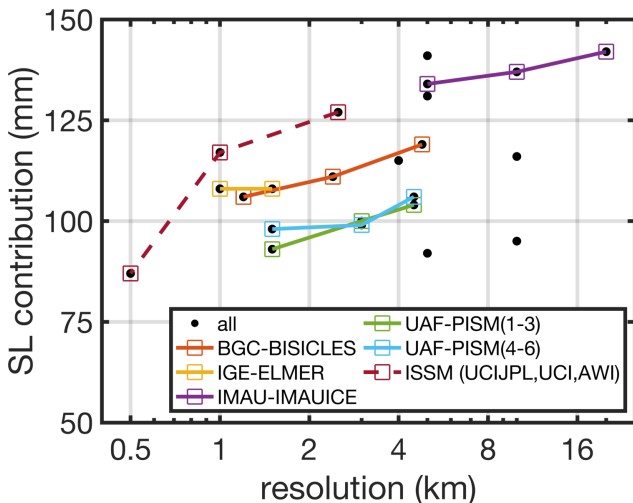

**Figure 1.** Results from the initMIP-Greenland exercise (Goelzer et al., 2018). Sea-level contribution versus (minimum) horizontal grid resolution of each participating ISM. Equal model versions but different grid resolutions are connected with a coloured line. ISSM model versions are connected with a coloured dashed line. Note the logarithmic scale of the x-axis. For unstructured meshes the finest resolution is displayed.

However, increasing the spatial resolution comes along with the ability to resolve the geometry and to track outlet glacier behaviour (Greve and Herzfeld, 2013; Aschwanden et al., 2016). Some previous works focused on the dependence of future mass loss of the GrIS on grid resolution (Greve and Herzfeld, 2013; Aschwanden et al., 2019). In these studies no clear conclusion how the resolution affects the mass loss was found. This was partly explained by the competing tendencies of SMB
and ice discharge that are differently resolved by the adopted resolutions. A separation of both responses in future projections experiments would elucidate how these two main sea-level contributors from the GrIS are affected by the horizontal resolution. Most likely coarse grids underestimate ice discharge as ice flow patterns and cross-sections of outlet glacier geometries are not well captured (Greve and Herzfeld, 2013; Aschwanden et al., 2016). High-resolution models, in turn, require a larger amount of computational resources. Unfortunately, when increasing the resolution, simple approximations to the momentum balance do
not provide an accurate solution (Pattyn et al., 2008). This limitation takes place particularly at the ice sheet margin and at outlet glaciers where all terms in the force balance become equally important (e.g. Pattyn and Durand, 2013). Due to the intensive computational resources needed to solve the full-Stokes equation, higher-order approximations provide a good compromise to balance model accuracy and computational costs on centennial time scales.

Determining whether increased model resolution is worth the extra computation time would be valuable to make progress
in narrowing uncertainties in ice sheet projections, even if only by a few per cent. The ISMIP6-projection-Greenland shows that models with low and high resolution are found at the upper and lower bound of sea-level contribution, though no specific analysis to the grid resolution is performed.

The main intention of this paper is to complement the study by Goelzer et al. (2020a) by evaluating the sensitivity of the simulated GrIS response to global warming due to different horizontal grid resolutions by one single ISM. Beside running the full scenarios (i.e. both oceanic and atmospheric forcing considered), we aim to explore the grid resolution dependence on atmospheric and ocean forcing separately. Therefore, the full scenarios are complement with experiments where either a changing SMB or the interaction of the glacier with the ocean is omitted. The simulations of the GrIS are performed with the Ice Sheet System Model (ISSM, Larour et al., 2012) and adopted spatial resolutions ranging from medium to high (4 and 0.75 km at fast-flowing outlet glaciers, respectively). The future scenarios build on climate forcing data from the CMIP5 global circulation model (GCM) MIROC5 under the Representative Concentration Pathway (RCP, Moss et al., 2010) 8.5 following the ISMIP6 protocol (Nowicki et al., 2020a).

## 2    Methods and experiments

Before presenting the concept of this study, we aim to address the terminology used for clarity. Following the glossary in Cogley et al. (2011), ice discharge is computed as the product of ice thickness $h$ and the depth-averaged velocity $\bar{v}$. In the following, the lower-case $q$ ($\mathrm{Gt\,a^{-1}\,m^2}$) refers to the local ice discharge at a point, and the upper-case $Q$ ($\mathrm{Gt\,a^{-1}}$) refers to the glacier-wide quantity (analogous for other quantities such as glacier-wide calving $D$ and local calving $d$). Quantities at the margin are reckoned in the normal direction.

### 2.1    Ice flow model ISSM

The model applied here is the Ice Sheet System Model (ISSM, Larour et al., 2012). It has been applied successfully to the GrIS in the past (Seroussi et al., 2013; Goelzer et al., 2018; Rückamp et al., 2018, 2019a) and is also used for studies of individual drainage basins of Greenland, e.g. the North East Greenland Ice Stream (Choi et al., 2017), Jakobshavn Isbræ (Bondzio et al., 2017) and Petermann Glacier (Rückamp et al., 2019b). ISSM is designed to use variable ice flow approximations ranging from shallow ice approximation to full-Stokes and has also the capability to perform inverse modelling to constrain unknown parameters.

Here, we make use of the Blatter-Pattyn approximation (Blatter, 1995; Pattyn, 2003) to obtain a most accurate solution even though computational time increased compared to simpler models (e.g Aschwanden et al., 2019). The system of equations are solved numerically with the finite element method and state variables are computed on each vertex of the mesh using piecewise-linear finite elements. The ice rheology is treated with a regularized Glen flow law (Glen, 1955), a temperature-dependent rate factor for cold ice, and a watercontent-dependent rate factor for temperate ice (Lliboutry and Duval, 1985).

At the ice base, sliding is allowed everywhere and the basal drag $\tau_b$ follows a linear viscous law (Weertman, 1957; Budd et al., 1984)

$$\tau_{b,i} = -k^2 N v_{b,i}, \tag{1}$$

where $v_{b,i}$ is the basal velocity vector in the horizontal plane and $i = x, y$. Although this type of friction law is often used in ice sheet modelling (Morlighem et al., 2010; Price et al., 2011; Gillet-Chaulet et al., 2012; Seroussi et al., 2013), it implies that the basal drag can increase without a bound. It was shown, that inducing an upper ratio of $\tau_b/N$ (Iken's bound) is more justified (Iken, 1981; Schoof, 2005; Gagliardini et al., 2007; Leguy et al., 2014; Joughin et al., 2019). However, we choose this type of friction law as it is commonly used in ISMs making use of inverse methods to constrain the basal friction (Morlighem et al., 2010; Larour et al., 2012; Seroussi et al., 2013; Perego et al., 2014; Gladstone et al., 2014).

The friction coefficient $k^2$ is assumed to cover bed properties such as bed roughness. The effective pressure is defined as $N = \varrho_i\, g\, h + \min(0, \varrho_w\, g\, z_b)$, where $h$ is the ice thickness, $z_b$ the glacier base and $\varrho_i = 910\,\mathrm{kg\,m^{-3}}$, $\varrho_w = 1028\,\mathrm{kg\,m^{-3}}$ the densities for ice and sea water, respectively. The parametrization accounts for full water-pressure support from the ocean wherever the ice sheet base is below sea-level, even far into its interior where such a drainage system may not exist. At marine-terminating glaciers water pressure is applied and zero pressure along land-terminating ice cliffs. A traction-free boundary condition is imposed at the ice/air interface.

The ISSM model domain for the Greenland ice sheet covers the present-day main ice sheet extent, and includes the current floating ice tongues (e.g., Petermann, Ryder and 79° North glaciers). The geometric input is BedMachine v3 (Morlighem et al., 2017). Thickness, bedrock and ice sheet mask is clipped to exclude glaciers and ice caps surrounding the main ice sheet. The initial ice sheet mask is manually retrieved from the data coverage of the MEaSURE velocity data set (Joughin et al., 2016, 2018) to ensure an available target for the employed basal friction inversion (sect. 2.3). A minimum ice thickness of 1 m is applied. Grounding line evolution is treated with a sub-grid parameterization scheme, which tracks the grounding line position within the element (Seroussi et al., 2014). A sub-grid parameterization on partially floating elements for basal melt is applied (Seroussi and Morlighem, 2018). The basal melt rate below floating tongues is parameterized with a Beckmann–Goosse relationship (Beckmann and Goosse, 2003). In this parameterization ocean temperature and salinity are set to $-1.7$°C and 35 Psu, respectively. The melt factor is roughly adjusted such that melting rates correspond to literature values (e.g. Wilson et al., 2017; Rückamp et al., 2019b). However, at most locations the grounding line coincides with the calving front. Except for the floating tongue glaciers Petermann, Ryder and 79° North, the sub-grid schemes at the grounding line is not applied. The treatment of the calving front evolution depends on the experimental setup and is explained in Sect. 2.4 and 2.5.2.

As we expect the grounding line not to retreat too excessively in the projections, model calculations with ISSM are performed on a horizontally unstructured grid which remains fixed in time. To limit the number of elements while maximizing the horizontal resolution in regions where physics demands higher accuracy, the horizontal mesh is generated with a higher resolution of $\mathrm{RES_{high}}$ in fast-flowing regions (observed ice velocity $> 200\,\mathrm{m\,a^{-1}}$) and a coarser resolution of $\mathrm{RES_{low}}$ in the interior. This adaptive strategy allows a variable resolution in key areas of the ice sheet, e.g. marine-terminating outlet glaciers. Experiments are carried out at four different horizontal grid resolutions with $\mathrm{RES_{high}}$ equal to 4, 2, 1, and 0.75 km (Table 1 and Fig. 2). The distribution of mesh vertices at numerous outlet glaciers is depicted in Figs. S6 to S19. In Figure 3, the interpolated bed elevation for two selected regions and grid resolutions is illustrated. Overall, the bedrock topography of the finer resolution shows deeper and fjord-like troughs which is closer to the BedMachine dataset.

**Table 1.** Summary of models and their mesh characteristics. Computational time is based on a projection run under MIROC5 RCP 8.5 and medium ocean forcing.

| Model name | RES$_{high}$ (km) | RES$_{low}$ (km) | number of elements | time step $\Delta t$ (yr) | computational time (minutes) | number of cores |
|---|---|---|---|---|---|---|
| G4000 | 4 | 7.5 | 1 169 546 | 0.100 | 83 | 90* |
| G2000 | 2 | 7.5 | 1 951 586 | 0.050 | 252 | 162* |
| G1000 | 1 | 7.5 | 4 241 020 | 0.025 | 640 | 342[†] |
| G750 | 0.75 | 7.5 | 6 220 928 | 0.010 | 1731 | 702[†] |

\* Intel Xeon Broadwell CPU E5-2697 v4, 2.3 GHz on the AWI HPC system Cray CS400.

[†] Intel Xeon Broadwell CPU E5-2695 v4, 2.1 GHz on the DKRZ HPC system Mistral.

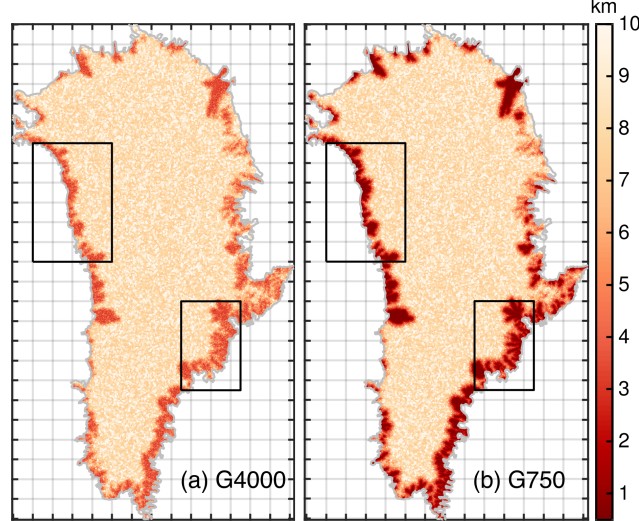

**Figure 2.** Horizontal mesh resolution (in km) used for G4000 (a) and G750 (b). Data are clipped at 0.5 and 10 km. The horizontal resolution of a triangle is defined by its minimum edge length. The grey line delineates the initial ice domain. Grey grid lines indicate 100 km. The black boxes indicate the northwestern and southeast subsets used in following figures.

Independent of the horizontal resolution, the vertical discretization comprises 15 terrain-following layers, refined towards the base where vertical shearing becomes more important. Please note, that G1000 and G750 correspond to the ISMIP6 contributions AWI-ISSM2 and AWI-ISSM3, respectively, in Goelzer et al. (2020a).

**2.2 Overview of experiments**

The ISMIP6 protocol requests the initialization mode prior or to the ISMIP6 projection start date. If the initialization date is before the start of the projections, a short historical run is needed to advance the ISM from the reference date to the end of

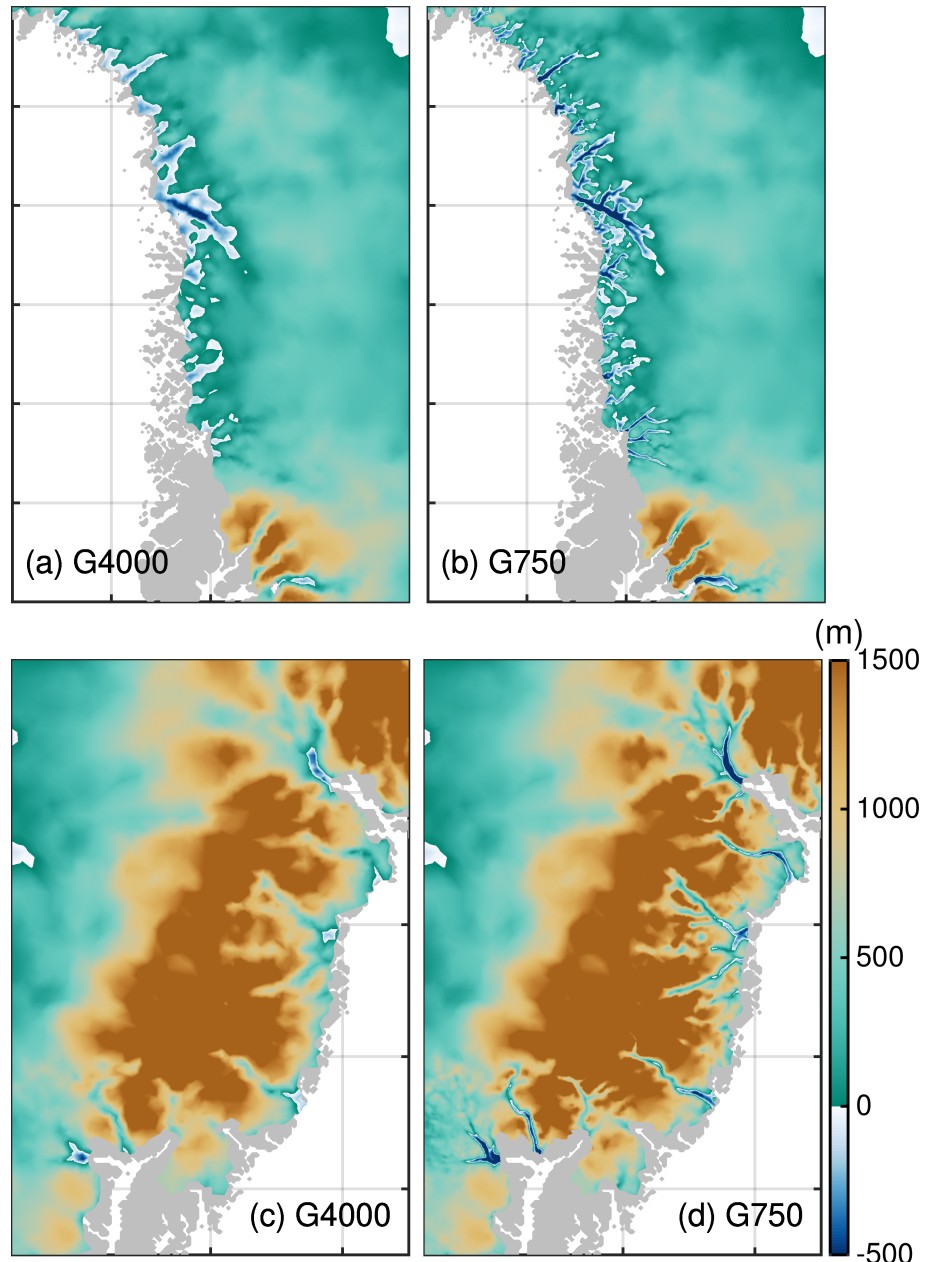

**Figure 3.** G4000 and G750 bedrock topography for the northwestern region (a,b) , respectively, and for the southeast region (c,d), respectively. Region subsets are shown in Fig. 2.

2014. From this date, the future climate scenarios branch off. Unforced constant-climate control experiments are defined to

capture the model drift with respect to the ISM reference climate and the ISMIP6 projection start date. The set of experiments are described in the following sections but can be summarized as follows:

- initialization: experiment to retrieve the initial state of the model.

- ctrl: experiment where the climate is held constant to the reference climate (from January 1991 to end of December 2100).

- ctrl_proj: experiment where the climate is held constant to the reference climate (from January 2015 to end of December 2100).

- historical: experiment to bring model from the initialization state to ISMIP6 projection start date (from January 1991 to end of December 2014).

- projection: future climate scenario (from January 2015 to end of December 2100).

## 2.3 Initialization experiment

The initialization state of ISSM is based on data assimilation and inversion for determining the basal friction coefficient. Before the inversion, a relaxation run assuming no sliding and a constant ice temperature of $-10°$C is performed to avoid spurious noise that arises from errors and biases in the data sets. To ensure that the relaxed geometry does not deviate too much from the observed geometry, the relaxation is conducted over one year. However, while inverse modelling is well established for estimating basal properties, the temperature field is difficult to constrain without performing an interglacial thermal spin-up. Therefore, we rely on a temperature field that was obtained by a hybrid approach between paleoclimatic thermal spin-up and basal friction inversion. This method was developed for the AWI contribution in initMIP-Greenland (Goelzer et al., 2018) and further improved in Rückamp et al. (2018) by using the geothermal flux pattern from Greve (2005, scenario hf-pmod2). Here, we initialize the ice rheology on the four employed G4000–G750 grids by interpolating the 3D temperature and watercontent fields from the hybrid spin-up in Rückamp et al. (2018). The basal melting rates of grounded ice are equivalently interpolated. During all transient runs, we neglect an evolution of the thermal field. This is justified as it was shown by Seroussi et al. (2013) and Goelzer et al. (2018, see submissions AWI-ISSM1 and 2) that the temperature field and its change has a negligible effect on century time-scale projections of the GrIS.

The main ingredient to the initialization is the inversion to infer the basal friction coefficient $k^2$ in Eq. 1. This approach minimizes a cost function that measures the misfit between observed and modelled horizontal velocities (Morlighem et al., 2010). The cost function is composed of two terms which fit the velocities in fast- and slow-moving areas. A third term is a Tikhonov regularization to avoid oscillations. The parameters for weighting the three contributions to the cost function are taken from Seroussi et al. (2013). We leverage horizontal surface velocities from the MEaSURE project (Joughin et al., 2016, 2018), as the data set with almost no gaps over GrIS is suitable for basal friction inversion.

The assigned reference year is 1990. This date is not in agreement with the timestamps of the BedMachine data set (reference time is 2007) and the MEaSURE velocity data set (temporal coverage from 2014 to 2018). However, we ignore the contempo-

**Table 2.** Summary of projection experiments based on MIROC5-RCP8.5 climate data.

| Experiment label | atmospheric forcing | oceanic forcing | combination |
|---|---|---|---|
| RCP8.5-Rlow | SMB anomaly | low | full |
| RCP8.5-Rmed | SMB anomaly | med | full |
| RCP8.5-Rhigh | SMB anomaly | high | full |
| RCP8.5-Rnone | SMB anomaly | - | atmospheric only |
| OO-Rmed | - | med | ocean only |
| OO-Rhigh | - | high | ocean only |

raneity requirement in the inversion approach and put more weight on to start the projections at the end of the assumed GrIS steady-state period (e.g. Ettema et al., 2009). All transient simulations start from the initial state, that means, we do not perform a subsequent relaxation run to bring the model to a steady-state (see sect. 2.6).

### 2.4 Historical and control experiments

In both control experiments (ctrl and ctrl_proj), the SMB and ice sheet mask remains unchanged to the reference year according to the ISMIP6 protocol. To advance the model from the reference time to the projection start date, the historical scenario is needed. During the historical period, yearly cumulative SMB is taken from the RACMO2.3p2 product (Noël et al., 2018) for the years from 1990 to 2015. For simplicity, the ice sheet extent remains unchanged to the reference year. This is a crude approach but representing the historical mass loss accurately was not a strong priority for our experimental setup. As the ice

front is not moving in these three scenarios ice discharge $Q$ equals calving $D$.

### 2.5 Future forcing experiments

It is beyond the scope of this paper to present the details of the ISMIP6 protocol and experimental design. Therefore, we aim to briefly outline the external forcing approach. Further details are given in Goelzer et al. (2020a), Nowicki et al. (2020a), Fettweis et al. (2020), and Slater et al. (2019, 2020).

As we aim to study the effect of grid resolution on ice mass changes, we run the future scenarios based on climate data from one single GCM. The GCM MIROC5 was selected as it performs well in the historical period and represents a plausible climate near Greenland (Barthel et al., 2020). The GCM output is used to separately derive ISM forcing for the interaction with the atmosphere and the ocean. We set up experiments where both external forcings are considered; these scenarios are termed as 'full' in the following (RCP8.5-Rlow/med/high). In addition, we perform simulations where either the atmospheric

forcing (RCP8.5-Rnone) or the marine-terminating outlet glacier retreat (OO-Rmed/high) is at play. The conducted projection experiments and the corresponding experiment labels used in this study are summarized in Table 2 and are explained in the following sections.

### 2.5.1 Atmospheric forcing

ISMIP6 provide surface forcing data sets for the GrIS based on CMIP GCM simulations. The GCM output is dynamically downscaled through the higher-resolution regional climate model (RCM) MAR v3.9 (Fettweis et al., 2017). The latter allows to capture narrow regions at the periphery of the Greenland ice sheet with large SMB gradients, which are likely not captured by the GCMs. The climatic SMB that is used as future climate forcing reads

$$\text{SMB}_{\text{clim}}(x,y,t) = \text{SMB}_{\text{ref}}(x,y) + \Delta\text{SMB}(x,y,t) + \text{SMB}_{\text{dyn}}(x,y,t), \tag{2}$$

with the anomaly defined as

$$\Delta\text{SMB}(x,y,t) = \text{SMB}(x,y,t)_{\text{GCM}-\text{MAR}} - \overline{\text{SMB}}(x,y)_{\text{GCM}-\text{MAR}}^{1960-1989}, \tag{3}$$

where $\text{SMB}(x,y,t)_{\text{GCM}-\text{MAR}}$ is the direct output of MAR using the GCM climate data and $\overline{\text{SMB}}(x,y)_{\text{GCM}-\text{MAR}}^{1960-1989}$ the corresponding mean value over the reference period (from January 1960 to December 1989). As the reference SMB field $\text{SMB}_{\text{ref}}(x,y)$, we choose the downscaled RACMO2.3p2 product (Noël et al., 2018) whereby a model output was averaged for the period 1960–1990. This period is chosen as the ice sheet is assumed close to steady-state in this period. (e.g. Ettema et al., 2009). The SMB deduced by MAR is processed on a fixed topography (off-line), consequently local climate feedback processes due to the evolving surface in the projection experiments are not captured. The SMB height-elevation feedback is considered with a dynamic correction $\text{SMB}_{\text{dyn}}$ to the $\text{SMB}_{\text{clim}}$ following Franco et al. (2012)

$$\text{SMB}_{\text{dyn}}(x,y,t) = \text{dSMBdz}(x,y,t) \times (z_s(x,y,t) - z_{\text{ref}}(x,y)). \tag{4}$$

The surface elevation changes are taken from the ISM elevation $z_s(x,y,t)$ while running the simulation and the corresponding ISM reference elevation $z_{\text{ref}}(x,y)$ from the initialization state. The yearly patterns of $\Delta\text{SMB}(x,y,t)$ and $\text{dSMBdz}(x,y,t)$ are provided by ISMIP6. A cumulative SMB anomaly over the projection period is shown in Fig. 4a.

### 2.5.2 Oceanic forcing

For the oceanic forcing we rely on the empirically derived outlet glacier parametrization retreat by Slater et al. (2019, 2020). This method circumvents the problem of employing a physically-based calving law and frontal melting rates based on GCM output. When employing this parameterization to the calving front, retreat and advance of marine-terminating outlet glaciers is directly prescribed as a yearly series of ice front positions. (i.e., is not a result of ice velocity at calving front, calving rate and frontal melt that is used to simulate the calving front position). Here, the binary retreat masks (i.e., ice and non-ice covered cells) are interpolated to the native grid by nearest neighbour interpolation. Retreat occurs once a cell is fully emptied.

Though this parameterization is a strong simplification, it builds on projected submarine melting taking into account changes in ocean temperature and surface meltwater runoff from a GCM. The parametrization is not applied to the individual glaciers but over a predefined geographical region. Based on the numerous retreat trajectories, a medium retreat scenario as the trajectory with the median retreat at 2100 is defined. To cover uncertainty by this approach, a low and high retreat scenarios is defined

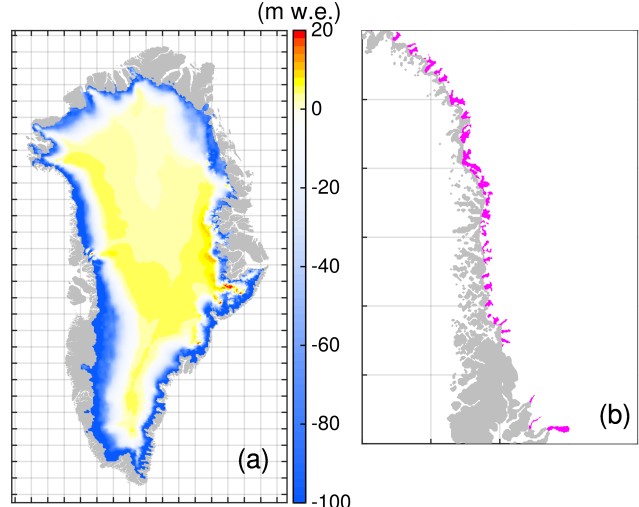

**Figure 4.** Atmospheric and oceanic forcing. (a) Spatial pattern of the cumulative (2015–2100) SMB anomaly based on MIROC5-RCP8.5 and downscaled with MAR (Fettweis et al., 2020). (b) Retreat of marine-terminating outlet glaciers in the northwestern region under RCP8.5-Rhigh scenario. Purple areas indicate retreated areas in 2100. Region subsets are shown in Fig. 2.

as the trajectories with the 25th and 75th percentile retreats at 2100. In the following, these retreat scenarios are termed Rlow, Rmed and Rhigh (Table 2). The retreat mask for RCP8.5-Rhigh in 2100 is exemplarily shown in Fig. 4b. Please note, that the

230 future projection experiment RCP8.5-Rnone experience no retreat of marine-terminating outlet glaciers.

## 2.6   Comparability of experiments

A central question about resolution dependence is always "How comparable are the results?" and "What is controlling the results?". The presented initialization procedure and involved parameters are achieved for the high-resolution simulations (G750). The simulations with a coarser resolution would probably require other parameters, e.g. to obtain a better result to

235 observational targets or to achieve a reduced model drift. However, we decided here to keep model parameters (e.g. inversion parameters) and parameterizations (e.g. sub-grid scheme at grounding line) unchanged for all simulations. Similarly for the retreat masks, we rely on binary retreat masks for all adopted resolutions although the ISMIP6 protocol requests a sub-grid scheme for coarse resolution models. On one hand this strategy simply assumes that the results are comparable as they build on the same basis. On the other hand it avoids exploring parameter spaces which are out of the scope of this study.

For the geometric input we are following the same strategy. It is always a compromise between matching the observed geometry or being closer to a steady-state. Here, we put more weight on having the initial geometry closer to the observed geometry. Therefore, we directly started the historical run after the inversion and no further relaxation run is performed to bring the model to a steady-state. As the model is likely not in steady-state at the initial state, we expect a model drift in the transient runs which would not be the case for models that do a relaxation towards a steady-state after the inversion.

## 3  Results

### 3.1  Historical scenario

To evaluate the modelling decisions pertaining the initialization, the state of the ice sheet at the end of the historical period is compared to observations. Due to the sparseness and limited temporal and spatial coverage of available observations, we rely on the BedMachine v3 (150 m grid spacing) and MEaSURE data sets (250 m grid spacing) for ice thickness and surface velocity, respectively. As these data are used in the data assimilation and inversion, velocity and thickness are not independent quantities. However, during the historical period the ice sheet state is altered by the boundary conditions and external forcing. Therefore, the following evaluation attempts to quantify differences from the model configurations at the ISMIP6 projection start date.

Since the results are qualitatively similar for each grid simulation (Figs. S1, S2 and S3), the surface velocity field of the G750 simulation is exemplarily shown in Fig. 5a. A consequence of the employed basal friction inversion is the high fidelity in simulating the observed velocity field indicated by a low root mean square error (RMSE) (Fig. 5b). Notable is the decreasing RMSE with increasing spatial resolution. At the end of the historical experiment the RMSE is increased compared to the initialization due to geometric and velocity adjustments over the course of the experiment. However, the ice sheet-wide RMSE of each model version is very similar but in the areas of fast-flowing outlet glaciers (observed velocity $> 200\,\mathrm{m\,a^{-1}}$) differences are more evident: The G4000 and G750 simulations yield $\mathrm{RMSE} = 150\,\mathrm{m\,a^{-1}}$ and $\mathrm{RMSE} = 80\,\mathrm{m\,a^{-1}}$, respectively. Note that these values are not identical to those given in Goelzer et al. (2020a), as the evaluation here is based on a different subsampling method. A mean signed difference (MSD) reflects a stronger underestimation of the simulated velocities with coarser resolution. The underestimation of prominent outlet glaciers for the G4000 setup is demonstrated in the spatial pattern of velocity differences (Fig. S4). With increasing resolution, the difference pattern becomes more heterogeneous. Although barely visible, the G750 setup provides an interesting signature at narrowly confined outlet glaciers: Generally, the velocities in the main trunk are underestimated while beneath the shear margin velocities are overestimated. This might be due to the fact, that the employed resolution is not able to resolve the sharp velocity jump across the shear margin.

A similar evaluation for the thickness is performed. The ice sheet-wide RMSE of ice thickness depicts the qualitative similar grid-dependent behaviour as the velocity evaluation (Fig. 5c) . Similarly, the RMSE show larger differences in the fast-flow regions: The G4000 and G750 simulations yield $\mathrm{RMSE} = 126\,\mathrm{m}$ and $\mathrm{RMSE} = 45\,\mathrm{m}$, respectively. In this region, the MSD indicates underestimation of ice thicknesses with coarser resolution. Spatial patterns of the thickness differences over the course of the historical experiment are shown in Fig. S5.

The stored volumes, ice extent and spatially integrated SMB is among all grid simulations rather similar ($V = 7.28\,\mathrm{m\,SLE} \pm 0.2\%$, $A = 1.787 \times 10^6 \mathrm{km^2} \pm 0.7\%$, $\mathrm{SMB} = 375\,\mathrm{Gt\,a^{-1}} \pm 0.2\%$). However, the underestimation of velocities and ice thicknesses in the coarser resolution models is confirmed by the temporal mean of the ice discharge in the historical period. The intrinsically simulated ice discharge $Q$ yields $207\,\mathrm{Gt\,a^{-1}}$ to $341\,\mathrm{Gt\,a^{-1}}$ for the G4000 and G750 simulations, respectively.

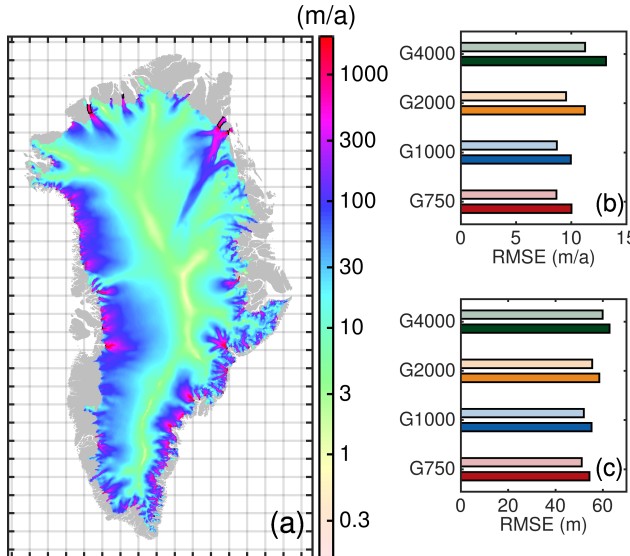

**Figure 5.** Simulation results and error estimate of model output at the end of the historical run compared to observations. (a) Simulated surface velocity of the GrIS (m a$^{-1}$) from the G750 simulation. The grey silhouette shows the Greenland land mask from BedMachine v3. Thin black lines show the grounding line. (b) Root mean square error (RMSE) of the horizontal velocity magnitude compared to MEaSURE. (c) RMSE of ice thickness (not corrected with ctrl) compared to BedMachine v3. In (b) and (c), light and dark colors represent diagnostics at the initialization and end of the historical experiment, respectively. The diagnostics have been calculated on the regular MEaSURE and BedMachine grids, respectively.

## 3.2 Sea-level contribution

In the following the transient effect of spatial resolution on ice volume evolution for the future-climate experiments is studied. The change in ice mass loss is expressed as sea-level contributions. Therefore, the simulated volume above flotation is converted into the total amount of global sea-level equivalent by assuming a constant ocean area of $3.618 \times 10^8 \, \mathrm{km}^2$. In the following, the mass losses in the projection experiments are corrected with the ctrl run with respect to the reference time. For all conducted projection experiments, the determined GrIS mass losses as a function of time are shown in Fig. 6 and listed in Table 3.

As we have not initialized our model to be at steady state the transient response in the ctrl experiment (thin coloured lines in Fig. 6) should not be interpreted as a prediction of actual future behaviour, the ctrl run rather confirms that each model has achieved a high degree of equilibration, which is reflected with a low rate of volume change. As the initialization states are presumably different across the employed grids, we expect a different response of the ice sheet as it is likely not in equilibrium with the applied SMB and ice flux divergence. The simulated ice mass evolution shows for all models a mass gain for the 111-year ctrl experiment ranging between -28 and -2 mm. With increasing resolution, the drift gets smaller and is minimal for the G1000 and G750 simulations. Although projections are corrected with the ctrl run, the higher drift needs caution when interpreting the results as it has, e.g. a consequence on the SMB height-elevation feedback. The higher mass gain rates of the

**Table 3.** Modelled mass change (mm SLE) in future experiments for all experiments.

| Experiment label | G4000 | G2000 | G1000 | G750 |
|---|---|---|---|---|
| **corrected with ctrl:**[†] | | | | |
| RCP8.5-Rlow | 118.3 | 119.7 | 118.6 | 118.7 |
| RCP8.5-Rmed | 122.5 | 125.8 | 126.4 | 126.5 |
| RCP8.5-Rhigh | 130.8 | 134.7 | 136.8 | 137.2 |
| RCP8.5-Rnone | 108.0 | 105.1 | 103.6 | 103.1 |
| OO-Rmed | 19.5 | 26.4 | 29.3 | 30.1 |
| OO-Rhigh | 28.9 | 36.7 | 41.4 | 42.6 |
| **corrected with ctrl_proj:**[†] | | | | |
| RCP8.5-Rlow | 115.8 | 117.6 | 117.1 | 117.2 |
| RCP8.5-Rmed | 120.0 | 123.7 | 124.8 | 125.1 |
| RCP8.5-Rhigh | 128.3 | 132.6 | 135.3 | 135.8 |
| RCP8.5-Rnone | 105.5 | 103.0 | 101.8 | 101.4 |
| OO-Rmed | 17.0 | 24.3 | 27.7 | 28.7 |
| OO-Rhigh | 26.5 | 34.6 | 39.9 | 41.1 |
| ctrl | -28.0 | -12.6 | -2.5 | -1.5 |
| ctrl_proj | -19.1 | -8.7 | -2.4 | -1.9 |

[†] Numbers for G1000 and G750 are different compared to Goelzer et al. (2020a) as they are differently calculated (e.g. considered ocean area, native versus interpolated grid resolution).

coarser resolutions in the ctrl simulation are due to the lower ice discharge rates (sect. 3.1). Although the integrated signal in ice mass change is generally small, the spatial patterns reveal an ice thickness imbalance up to hundreds of metres over the ctrl period (Fig. 7). Imposing a SMB correction to suppress the thickness imbalance would be feasible for maintaining a small drift. However, this is avoided here to enable a clean comparison between the four model version and to leave the ice dynamics some degree of freedom. Moreover, the mass trends represent an important diagnostic. Comparing the ice thickness changes reveal distinct differences between the grid-resolution simulations (Fig. 7). For example, at the end of the ctrl run, at some western and north-western locations at the margin the G4000 simulation exhibit thickening while the G750 reveals thinning. Another example is simulated at the south-western margin, where extensive thickening is prevailing in all simulations but reaches farther inland in the coarser resolutions. However, from these figures, it becomes clear that positive and negative thickness changes partially compensate, resulting in a low model drift.

Depending on the projection scenario, the GrIS will lose ice corresponding to a SLE between 19 mm (or 108 excluding OO-Rmed/high) and 137 mm. For the future climate scenarios including atmospheric forcing a gradual increase in mass loss until the end of this century is simulated, indicating accelerating mass loss for a high-emission scenario. For the RCP8.5-

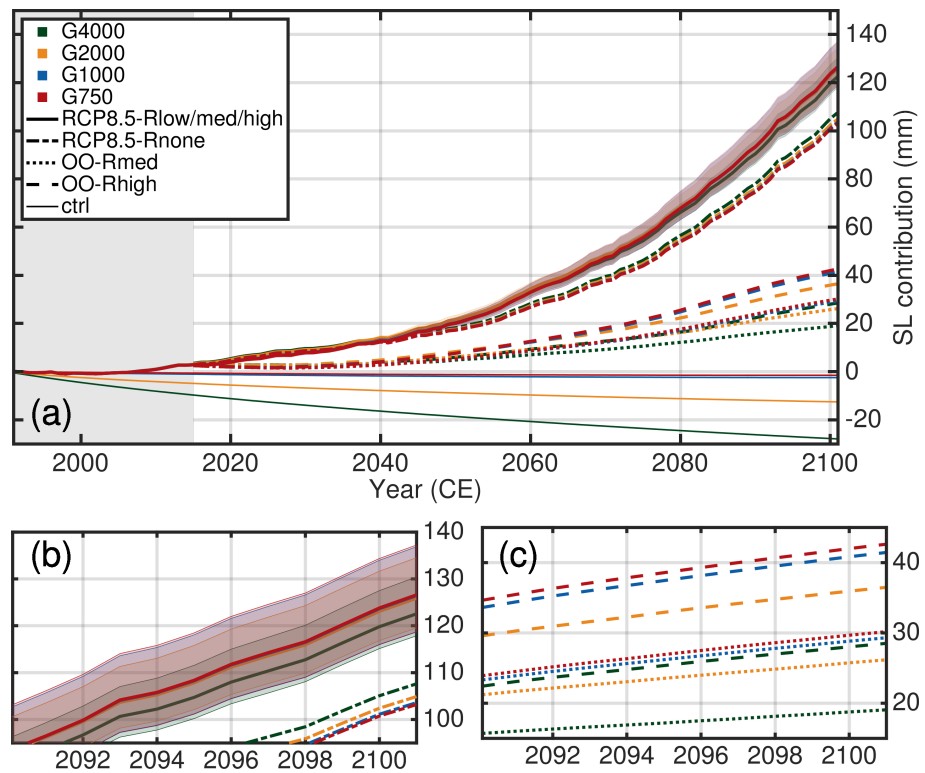

**Figure 6.** Projected sea-level contribution of the Greenland ice sheet based on MIROC5 RCP 8.5 climate data (a). Coloured lines indicate the different employed grid-resolutions while the individual scenarios are indicated with different line styles. The mass loss trends are corrected with the ctrl run relative to the reference time. The grey shaded box shows the historical period. (b) Zoom to the RCP8.5-Rlow/med/high/none scenarios. (c) Zoom to the OO-Rmed/high scenarios.

Rmed the mass loss reaches about 125.3 mm in 2100 (mean over G4000, G2000, G1000 and G750 results). The uncertainty
quantification in the oceanic forcing results in a mean sea-level contribution, that is 7.1% less and 5.4% greater for the RCP8.5-
Rlow and RCP8.5-Rhigh scenarios, respectively. When no calving front retreat is at play, i.e. the RCP8.5-Rnone scenario, the
projected mean mass loss is approx. 105.0 mm, i.e ~20 mm less compared to RCP8.5-Rmed. In contrast, the mean mass loss
is considerably reduced to 26 mm and 37 mm in the OO-Rmed and OO-Rhigh experiment, respectively. Interestingly, a linear
superposition of RCP8.5-Rnone and OO-Rmed leads to an overestimated mass loss of about 4.1% for G4000 and 5.3% for
G750 compared to RCP8.5-Rmed where both external forcings are simultaneously at play; a linear superposition of RCP8.5-
Rnone and OO-Rhigh leads to 4.5% and 5.8% overestimation. This is inline with earlier studies where this effect was already
reported (Goelzer et al., 2013; Fürst et al., 2015)

Among all future projections a resolution-dependent impact on sea-level contribution is generally small compared to the
total signal for our grids. In 2100, the spread in sea-level contribution is 6.4 mm in RCP8.5-Rhigh, 4.1 mm in RCP8.5-Rmed,
1.5 mm in RCP8.5-Rlow and 5 mm in RCP8.5-Rnone. Merely the OO-Rlow/med scenarios exhibits a spread of 10.7 mm and

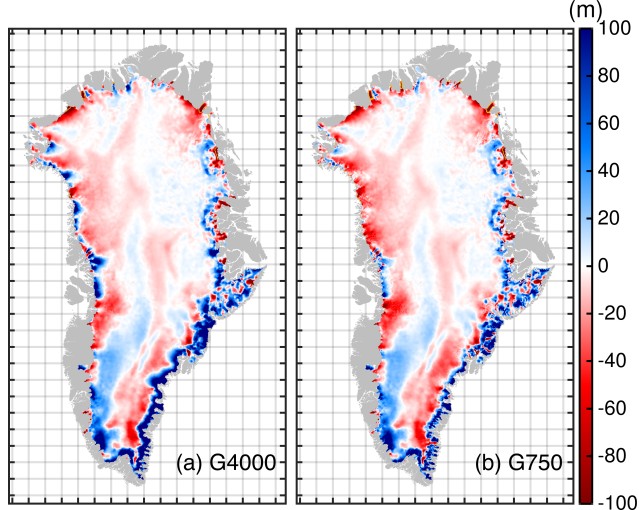

**Figure 7.** Difference of ice thickness between 2100 and 2015 for the ctrl run. (a) G4000 simulation and (b) G750 simulation. The grey silhouette shows the Greenland land mask from BedMachine v3. Positive values represents thickening, and negative shows thinning. Thin yellow line show the grounding line at the year 2100.

13.6 mm, respectively, which is in the order of the absolute magnitude. A notable feature for all conducted simulations is, that the sea-level contribution in each individual experiment converges with increasing resolution.

Figure 8 summarizes the qualitative behaviour of each experiment as function of grid resolution. Note that the sea-level contribution in each experiment is normalized by its maximum. The finer resolutions tend to produce more mass loss in 2100 for the RCP8.5-Rmed/high and OO-Rmed/high experiments. An inverse behaviour is determined for the RCP8.5-Rnone experiment. The trend in the RCP8.5-Rlow experiment is not clear. The RCP8.5-Rnone and OO-Rmed/high experiments unveil a linear behaviour as a function of grid size with regression slopes of $m = 1.50 \, \text{mm} \, \text{km}^{-1}$, $m = -3.27 \, \text{mm} \, \text{km}^{-1}$, and $m = -4.18 \, \text{mm} \, \text{km}^{-1}$ respectively. The trend in the full RCP8.5-Rlow/med/high scenarios is not consistent: RCP8.5-Rmed and RCP8.5-Rhigh show a peak in mass loss at the finest resolution, whereby a peak in mass loss is attained in the G2000 simulation for RCP8.5-Rlow. For the latter, it is worth to mention that the variations across the different grid simulations are less than 1.2%. However, an intriguing effect of the conducted simulations remains the opposite behaviour of the RCP8.5-Rnone and e.g. RCP8.5-Rhigh scenarios. In the following section, we study this effect by analysing the mass partition to get a more in-depth insight into the role of atmospheric and oceanic forcing on grid-resolution. It is worth to mention that the qualitative behaviour of the detected grid-dependent mass loss remains similar when the projections are corrected with the ctrl_proj experiment (Table 3).

### 3.3 Mass partitioning

The relative mass loss partitioning in 2100 is shown in Fig. 9 to explore the role of the grid resolution in each experiment. The bars indicate the relative importance to sea-level contribution of ice dynamic changes in the projections. The dynamic

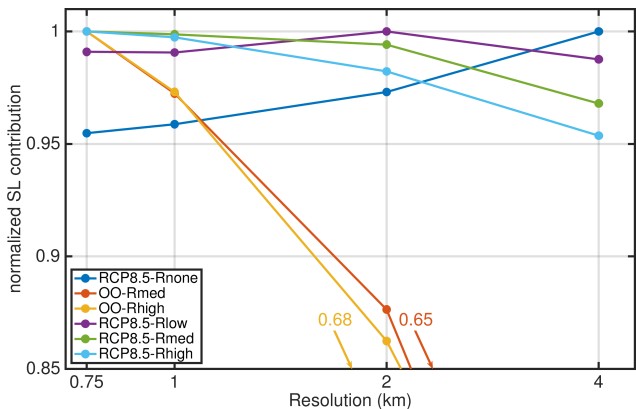

**Figure 8.** Projected sea-level contribution in 2100 of the Greenland ice sheet as function of the horizontal grid size. Values are normalized to the maximum of each experiment (coloured lines). Note the logarithmic scale of the x-axis.

contribution (composition of front retreat and ice discharge) is calculated as the residual of the total mass change and the
integrated SMB anomaly. The remainder explains the part of SMB. The overall picture reveals that for experiments that include
the atmospheric forcing the SMB anomaly is the governing forcing regardless of the grid resolutions. However, the importance
of the dynamic contribution increases with larger prescribed retreat rates of outlet glaciers; i.e. G750 with RCP8.5-Rhigh on
the upper end shows the highest importance of dynamic contribution with up to ∼28.4%. On the lower end, the RCP8.5-Rnone
shows diminished importance of dynamic contribution (<5%). In the OO-Rmed/high scenarios, the mass loss is dominated
by dynamic contribution. Concerning the grid resolution, the importance is on an equal level and exceeds 100%. The negative
importance of SMB stems from the fact that the glacier retreat is cutting off regions at the ice sheet margin where the static
SMB is low.

In the full experiments RCP8.5-Rlow/med/high, an increase in resolution enhances the importance of dynamic contribution.
For the G750 simulation it is ∼3, 5 and 6% higher for RCP8.5-Rlow/med/high, respectively, compared to G4000. Curiously, the
opposite behaviour is observed for the RCP8.5-Rnone experiment, where a finer resolution damps the importance of dynamic
contribution; G4000 yield 4.9% whereby G750 2.9% dynamic contribution.

The simulated inverse grid-resolution responses raise the question of the driving causes. Overall the time series of the SMB
show a decline and only minor differences among the grid resolutions (Fig. 10a). At the end of the projection, the cumulative
SMB is 2.1% and 2.6% lower in the G4000 simulation for RCP8.5-Rnone and RCP8.5-Rhigh, respectively, compared to G750.
These differences could be explained by different evolution of ablation areas at the margin and the SMB height-elevation
feedback, in particular, affected by the ctrl run, among all grid-resolution setups. In contrast, the cumulative ice discharge for
these settings reveals an opposing response in the RCP8.5-Rnone and RCP8.5-Rhigh scenarios and more relative differences
between the grid resolutions (Fig. 10b and c). At least for G2000, G1000, and G750, the ice discharge in the RCP8.5-Rnone
experiment decreases over the century; the decrease in G4000 is offset by a few decades and exhibits an increase early in
the century. These reductions explains the grid-dependence of the dynamic contribution as listed in the previous paragraph

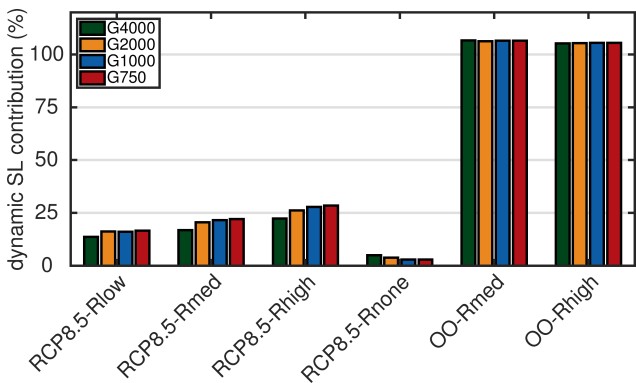

**Figure 9.** Mass loss partitioning for the conducted experiments. The bars indicate the relative dynamic contribution to sea-level, calculated as the residual of total the mass change and the integrated SMB anomaly. The residual is a composition of front retreat and ice discharge.

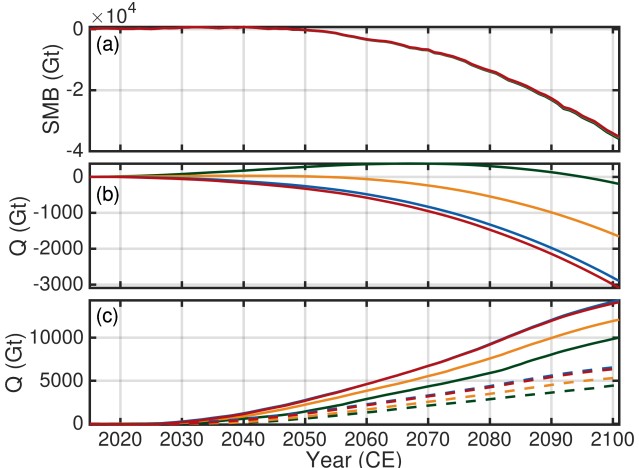

**Figure 10.** Time series of cumulative SMB anomaly and cumulative ice discharge $Q$. Colour scheme is as in previous figures. (a) Cumulative SMB anomaly for RCP8.5-Rnone. (b) Cumulative ice discharge $Q$ for RCP8.5-Rnone. (c) Ice discharge for RCP8.5-Rhigh (solid lines) and RCP8.5-Rlow (dashed lines). Cumulative SMB anomaly for RCP8.5-Rhigh and RCP8.5-Rlow is qualitatively similar to RCP8.5-Rnone. Ice discharge is not corrected with the ctrl run.

(RCP8.5-Rnone in Fig. 9). For RCP8.5-Rhigh, the ice discharge shows an increase consistently but is more enhanced in the finer resolutions. This finding corroborates with the grid-dependent increase of the relative ice discharge importance (RCP8.5-Rhigh in Fig. 9). As the opposing differences in RCP8.5-Rnone and RCP8.5-Rhigh are prevailing in ice discharge, it can be concluded that resolving ice discharge on the different grids is a decisive factor here. The involved feedback are further
explored by focusing on particular outlet glaciers in the next section.

## 3.4 Outlet glacier response

The fact that the centennial mass loss for the full experiments increases as the grid size reduces raises the question whether this is caused by ice dynamics alone, dominant feedback with surface mass balance or the retreat, or other non obvious factors. We conduct an in-depth analysis of numerous prominent outlet glaciers at GrIS (Fig. S6 and table with analysis provided as separate SI). The responses of most of the outlet glacier reveal the deduced grid-dependent behaviour where higher resolutions cause an enhanced discharge. This is exemplary illustrated in Fig. 11a for Helheim Glacier. However, this behaviour could not be adopted to all selected outlet glaciers. The presented example demonstrates that the bedrock topography deviates significantly among the different grid-resolutions. Generally, the bedrock topography of the coarser resolution is located above the bed from the finer resolution. This topographic effect is restricted to narrow confined outlet glaciers that obey a characteristic width in the order of a few kilometres. Outlet glaciers that have a larger characteristic width, such as Humboldt glacier, reveal in our setups a comparable bedrock topography. Theses glaciers seem to have a qualitatively equal behaviour for glacier speed-up and change in ice discharge for all employed grid resolutions (Fig. 11b). This analysis demonstrates that adjacent glaciers that experience similar environmental conditions may behave differently because ice discharge is strongly controlled by glacier geometry.

Glaciers that are converted from a marine-terminating to a land-terminating glacier by retreating out of the water build an own class. These glaciers are no longer subject of the retreat and show a collapse in ice discharge regardless of the grid resolution as illustrated for Store Glacier in Fig. 11c. The qualitative behaviour of the retreat seems to be similar as reported in Aschwanden et al. (2019, Fig. 4b therein), but the timing of the retreat is different. In our study, Store Glacier is unstable and retreats within this century out of the water, while in Aschwanden et al. (2019) Store Glacier is in a very stable position; the quick retreat sets in far beyond 2100 once the glacier loses contact with the bedrock high. This different response is related to the employed retreat parametrization that lacks information of the bedrock topography, such as topographic highs and lows.

The RCP8.5-Rnone shows a distinct slow-down in ice velocities as illustrated in Fig. 11d for Store glacier. Visible is a larger slow-down of the higher resolutions; the same behaviour holds for the ice discharge $q$. This is in-line with the finding above, that the scenario RCP8.5-Rnone reveals reduced ice discharge (Fig. 10b).

## 4 Discussion

The simulations presented here show that the projected sea-level contribution is sensitive to the spatial resolution. The sensitivity effect depends on the climate forcing, with oceanic and atmospheric forcings showing opposite and non-trivial responses. The simulations have turned out that the ice discharge to the ocean is a decisive factor here controlling the grid-dependent spread. As shown above, outlet glaciers respond differently to external forcing, dependent on the employed grids and geometrical setting. In such a non-linear system examining a driving mechanism remains challenging. However, despite the somewhat heterogeneous response of outlet glaciers the different scenarios tend to produce an overall trend in characteristic fields that explains the different responses.

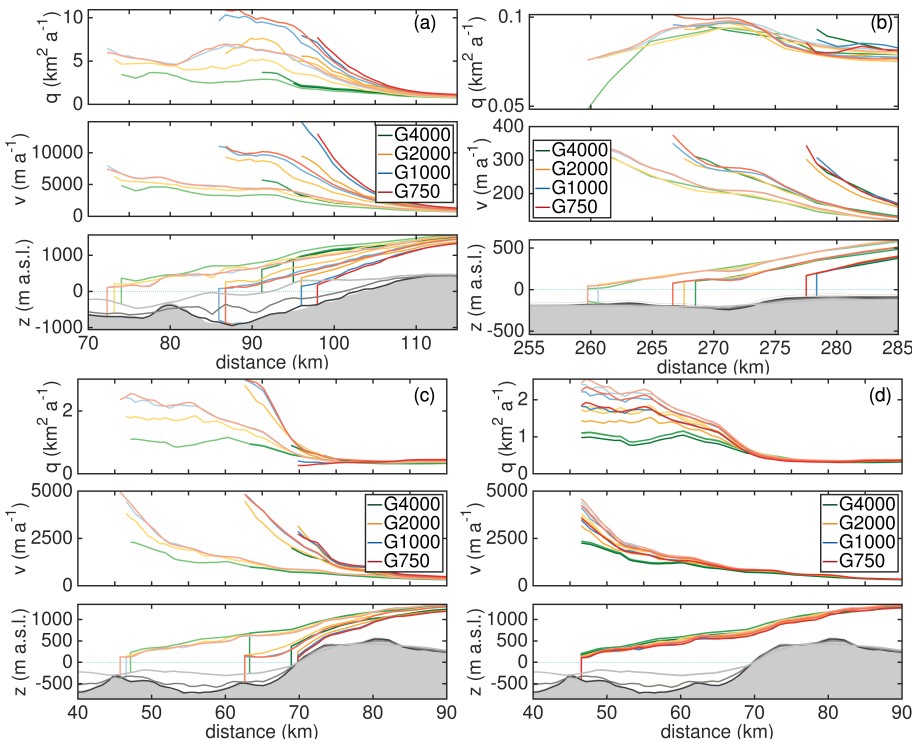

**Figure 11.** Response of outlet glaciers. Colour scheme is the same as in previous figures and light to dark colours indicate the years 2015, 2070 and 2100. (a) Helheim Glacier under RCP8.5-Rhigh forcing. (b) Humboldt Glacier under RCP8.5-Rhigh forcing. (c) Store Glacier under RCP8.5-Rhigh forcing. (d) Store Glacier under RCP8.5-Rnone forcing. Upper rows show the transient behaviour of the ice discharge $q$, the middle rows the surface velocity $v$ and lower rows the evolution of the ice geometry. In the lower rows, the grey shaded area depicts the bedrock topography from the G750 simulation. The grey lines from dark to light indicate the bedrock topography from the G1000, G2000 and G4000 simulation. None of the quantities are corrected with the ctrl run. Distance is relative to an arbitrary point.

The different responses in the full scenarios could be attributed to the ability to resolve bedrock topography and the interaction with basal sliding. Figure 12 illustrates spatial changes in the effective pressure and basal sliding velocity for RCP8.5-
395 Rhigh. A common characteristic for G750 is a stronger decrease in effective pressure, which is concentrated in areas where the finer grid shows a deeper bed of the marine portions compared to G4000. Due to the linear dependence of $\tau_b$ on effective pressure (Eq. 1), basal sliding velocities increase stronger in the finer resolution. This feedback is enhanced as the SMB perturbations lead to a decrease in ice thickness, hence in a decrease of the effective pressure. The transient evolution reveals further that thinning and acceleration propagate faster and farther upstream in the finer resolution. The higher signal propaga-
400 tion rates may have additional consequences on longer time scales as the surface melt is amplified by the positive surface mass balance-elevation feedback exposing the ice surface to higher air temperatures.

It remains questionable, if the widespread glacier acceleration is induced by the frontal stress perturbation instead of the decrease in the effective pressure. To isolate this effect we conduct a RCP8.5-Rhigh simulation (not shown) where the effective

pressure is held constant to the historical level. This setup reveals a very limited acceleration of a few glaciers in the G4000 simulation; some show no response or even a slow-down. In the corresponding G750 simulation most of the outlet glaciers show a speed-up but this effect is very localized and do not reach far upstream. Therefore, we conclude, that the pronounced decrease of the effective pressure along with the acceleration of outlet glacier is a dominant mechanism controlling the grid-dependent spread.

In order to investigate whether the response behaviour is an effect by purely reducing the grid size, we repeated the OO-Rhigh and RCP8.5-Rhigh experiments with a G1000 simulation using re-gridded bedrock topography and friction coefficient from the G4000 initial state (simulations are not shown). This setup adopts a high-resolution grid but omits detailed information from the high-resolution input data. Projected mass loss by this setups is closer to the G4000 simulation. They, therefore, demonstrate that a high model resolution alone is insufficient to explain the grid-dependent sea-level contribution. As a consequence, a driving cause for the grid-dependent behaviour arises from additional information in the input data. Therefore, we conclude that the grid-dependent behaviour is highly connected to the bedrock topography because the different models represent the bedrock topography quite differently.

Compared to the full and ocean only scenario, the atmospheric only scenario unveils a stronger mass loss for the coarse resolution. To some extent, this could be explained by a slightly lower SMB in the coarser resolution. However, the finer resolution produce a stronger reduction of ice discharge over the course of the experiment. Although for many of the outlet glaciers the effective pressure decreases (not as strong as for the scenarios with considered retreat), there is instead a slowdown of most glaciers (Fig. 13c,d). Again, these differences are concentrated in areas where the finer grid shows deeper troughs. Curiously, the finer resolution is better able to resolve these details, but the velocity evolution causes an extra reduction in ice discharge compared to coarser resolution. This non-trivially response is illustrated with the change in driving stress, approximated as $\tau_d = \varrho_i g h |\mathrm{grad} z_s|$ (Fig. 13a,b). Compared to 2015, the driving stress has locally decreased more in the finer resolution in 2100 compared to the coarser resolution; away from the marginal region, the driving stress changes are on a comparable magnitude for all employed grids. These results indicate that the reduction of the effective pressure is outperformed by geometric adjustments in the RCP8.5-Rnone scenario. This experiment intended to omit an interaction of the glacier with a changing ocean forcing, but the assumption of a fixed calving front hinder outlet glaciers from adjusting freely to topographic changes. They, therefore, do not experience reduced buttressing or frontal stress perturbations which are necessary mechanisms to trigger widespread glacier acceleration (e.g. Bondzio et al., 2017). In future studies, it might be desirable to allow the calving front to adjust although the oceanic forcing is held constant. Nevertheless, the simulations indicate that without a frontal stress perturbation an ensuing speed-up of the outlet glacier is not initiated. This highlights the importance for capturing calving events, i.e tracking the ice front position in numerical models, most accurate.

The inverse grid-dependent behaviour of the RCP8.5-Rnone and OO-Rmed/high scenarios have some implications when interpreting the mass loss of the ice sheet. The combined scenarios demonstrate that in a particular case, the sea-level contribution is maximized for an intermediate resolution. Depending on the horizontal grid resolution, the competing tendencies of SMB and ice discharge are differently resolved. This finding seems to corroborate with results by Aschwanden et al. (2019), where an intermediate resolution reveals the largest sea-level contribution.

A convergence of the grid-dependent estimates of sea-level contribution emerges around $\text{RES}_{high} \leq 1\,\text{km}$. This value corroborates with Aschwanden et al. (2016) for capturing outlet glacier behaviour indicating an upper limit for horizontal grid resolution. However, the converging behaviour should be treated with some caution. We cannot exclude whether a model resolution finer than $750\,\text{m}$ would lead to results that deviate from the convergence. On the one hand, the $150\,\text{m}$ horizontal grid spacing of the BedMachine v3 data set is much finer than our finest resolution of $750\,\text{m}$. As the retreat parametrization is insensitive to bed undulations, resolving the outlet glacier cross-sections is important for accurately model ice discharge. Since the glacier cross-sections are reasonable well approximated in G750, we do not expect that a resolving the geometry higher would drastically alter ice discharge rates. On the other hand, there are indications that at a resolution of $750\,\text{m}$ the HO solution is not fully converged (Pattyn et al., 2008). Adopting a higher resolution could have implications for the ice flow, and hence for the evolution of ice discharge. Likewise, in Aschwanden et al. (2019, Fig. S4 therein), the ice discharge is shown to increase as the mesh resolution is increased, and seem to converge below a resolution of $\leq 1800\,\text{m}$. However, the finer resolutions of $450$ and $600\,\text{m}$ seem to produce again a somewhat lower ice discharge. That might indicate that the underlying processes are not fully converged and still causing changes in mass loss trends.

Our grid-dependent results under atmospheric only forcing correlates with the finding in Goelzer et al. (2018, Fig. 1) and the Exp. C2 in Greve and Herzfeld (2013, Figs. 7a and b therein). Interestingly, the causes for the same behaviour seem to have different origins. In Goelzer et al. (2018) the effect is likely due to an overestimated ablation area (see also Goelzer et al., 2020b), whereby in our study the effect is attributed to the change is ice discharge of the ice sheet. The cause for the grid-dependent behaviour in Greve and Herzfeld (2013) is not specified further. Still, it is worth mentioning that they report a much better agreement of simulated to observed surface velocities by increasing the spatial resolution. Our experiments with the considered retreat of outlet glaciers could not be compared to the scenarios S1, M2 and R8 experiments in Greve and Herzfeld (2013). On the one hand, the external forcing approach differs. On the other hand, a grid-dependent behaviour in Greve and Herzfeld (2013) is not clear (except for the enhanced sliding experiment S1, where the finer resolution setups show a higher response). However, please note that the comparison to Aschwanden et al. (2019) and Greve and Herzfeld (2013) is limited as the studies employ different flow approximations and how basal friction is treated. The underlying physics in each ISM probably depends differently on the resolution.

Besides the fixed calving front in the atmospheric only scenario, further limitations of our study must be noted. The spread of projected sea-level contribution among all grids is likely subject to the chosen type of friction law. The choice of the basal friction law used in ISMs remains a matter of debate (Stearns and van der Veen, 2018; Minchew et al., 2019) and a potential source of uncertainty in sea-level projections (e.g. Brondex et al., 2019). Compared to our used type of friction law, there are some indications that friction laws satisfying an upper bound for the basal drag are more reliable (e.g. Schoof, 2005; Gagliardini et al., 2007; Leguy et al., 2014; Joughin et al., 2019). In brief, these type of friction laws invoke a switch for the friction regimes (low and high $N$, respectively) so that the influence of the effective pressure on the basal drag at slow ice flow is vanishing. It would be most interesting to evaluate their sensitivity to the horizontal resolution for projections on centennial time scales.

Another limitation concerns the choice of inversion parameters. We performed the inversions for basal parameters for each grid resolution individually but relying on the inversion parameters tuned for the high-resolution setups. Effectively, this re-

sults in an overall comparable pattern for the flow velocities (Fig. S1 and S2) and basal friction coefficient (Fig. S3) for all grids. However, on smaller scales, the inversion approach produces significantly different $k^2$ in many glacier basins. Recalling the relationship between $N$ and $k^2$, these different patterns are plausible but could potentially be a result from non optimal inversion parameters. However, this different spatial patterns might an additional contribution to the grid dependence of the simulations. In future studies, it will be worth investigating this influence only, e.g. by tuning the inversion parameters for each grid separately to find the optimal parameters.

However, the simulations conducted here reveal a grid-dependent spread in the full scenarios ranging between 1.2 and 5.3%, which is of comparable magnitude as the surface mass balance-elevation feedback (Eq. 4). The latter is recognized as an important mechanism and accounts for an additional sea-level contribution of about 6-8% (Goelzer et al., 2020a). A feedback that we have not considered is the enhanced surface melt influencing the basal conditions and calving by filling up crevasses. Given that we greatly simplify the representation of $N$, the effect of a reorganization of basal conditions (in either way) or the effect of increased availability of water due to increasing surface melt on basal sliding is suppressed. To overcome this limitation, an adequate subglacial hydrology model could be invoked, even if not considering seasonality. Subglacial hydrology model has shown the localized effect on $N$, which is likely having consequences on the spread between the employed grid resolutions (e.g. Werder et al., 2013; de Fleurian et al., 2016; Rathmann et al., 2017; Sommers et al., 2018; Beyer et al., 2018; Neckel et al., 2020).

A feedback that is not fully covered in our simulations is shear margin weakening and its influence on the evolution of flow velocities. Although the shear margins are weakly developed in the simulations (more pronounced in the finer resolutions), it is expected that a thermo-mechanical coupling could further weaken the shear margins as a response to frontal stress perturbations (e.g. Bondzio et al., 2017). Such a coupling would increases the widespread inland flow acceleration and enhances the rate of mass loss. However, the change in $\tau_b$ and $\tau_d$ is very pronounced in and around the main trunks and quite differently for the adopted grids (see Fig. S20 and 21). These patterns exemplify the need for resolving the shear margins particularly high, which we have not fully accomplished in this study as shear margins are becoming, in numerous cases, sub-grid phenomena. This may be a reason for under-representing glacier velocities inside the main trunk and over-estimate velocities outside the main flow as apparent in G750. This effect shall be addressed in further studies, in which ideally higher resolutions are employed or error estimators are engaged (e.g. dos Santos et al., 2019).

## 5   Conclusions

We applied the three-dimensional finite-element higher-order model ISSM to the Greenland ice sheet to simulate the future response under climatic changes specified by the ISMIP6 protocol. The sensitivity of mass changes to the spatial resolution is tested by employing four different grids with varying horizontal resolution ranging from 4 to 0.75 km at fast-flowing outlet glaciers. The simulations reveal up to ∼5.3% more sea-level rise compared to the coarser resolution in the full scenario RCP8.5-Rhigh and ∼3.2% for RCP8.5-Rmed. In scenarios where a change in SMB is omitted, and only outlet glacier retreat is at play, the finer resolutions produce significantly more mass loss (up to 33%). When no retreat is enforced, the sensitivity of the

grid-dependence exhibits an inverse behaviour, i.e. the coarser resolutions produce more mass loss. This finding is important to recognise for ice sheet models that have SMB as the dominant mass loss driver.

The results presented underline the importance of resolving the bedrock topography accurately. Areas with simple and low bedrock undulations experience a similar response in all model resolution. In areas with complex and high bedrock undulations striking differences between the employed grids emerge. A mechanism that exerts an important control on the resolution dependent spread is basal sliding predominantly in marine portions of outlet glaciers glacier. Since we rely on a greatly simplified effective pressure parametrization, further work with is needed to prove the robustness of this conclusion.

Given the strong interaction of the bedrock topography with sliding, it is obvious, that the major outlet glacier should be surveyed with the latest radar technology to obtain a substantially improved survey of the bedrock topography, the area of expected retreat and connected areas further upstream. This, in turn, requires ice sheet models ready to resolve these areas in grids and physics adequately.

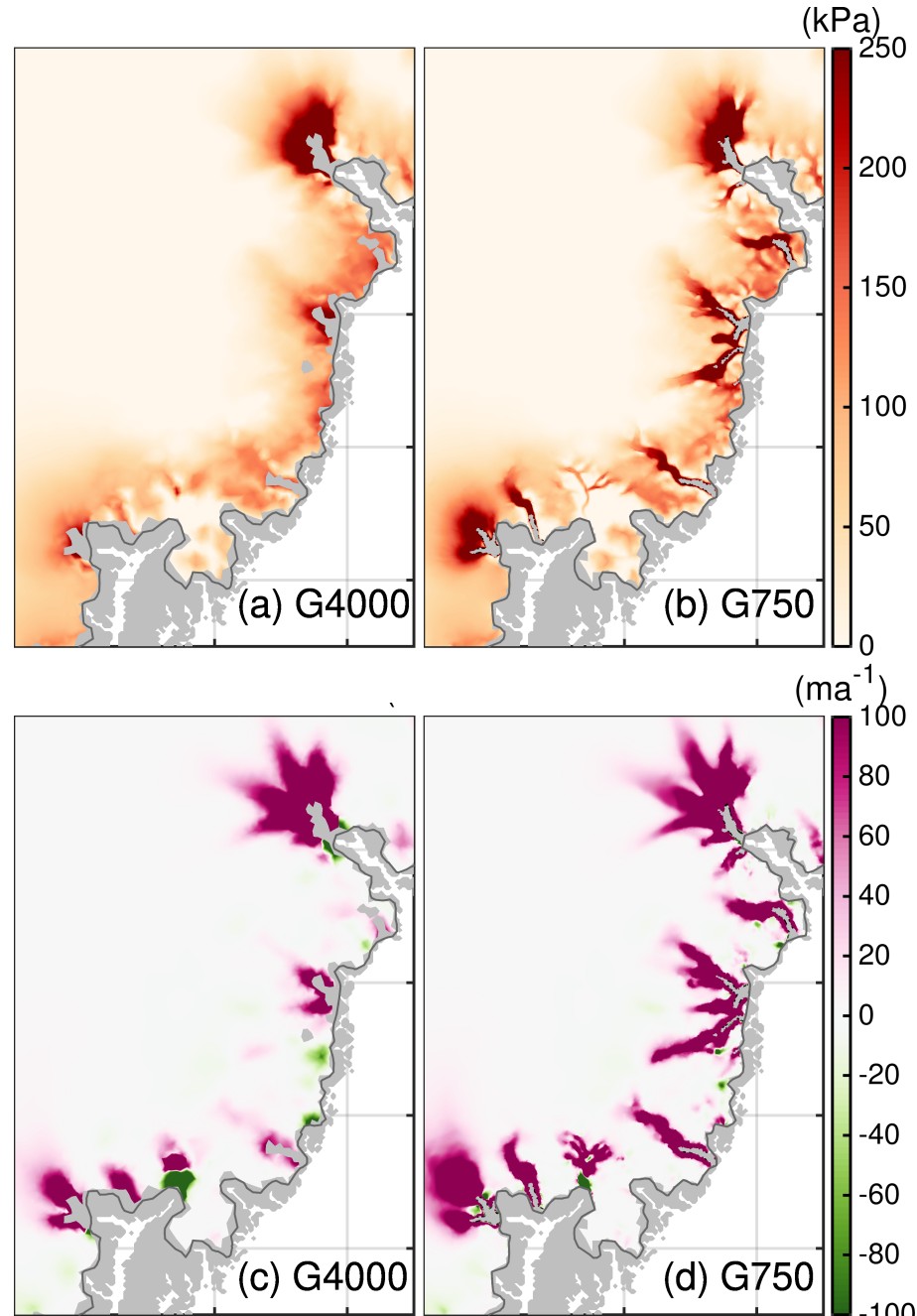

**Figure 12.** (a,b) Difference of effective pressure between 2100 and 2015 for the southeast region. (c,d) Difference of basal velocity between 2100 and 2015 for the southeast region. Region subsets are shown in Fig. 2. Dark gray line indicates the initial ice extent. Thin black line indicating the grounding line is not visible as it falls together with the calving front. The grey silhouette shows the Greenland land mask from BedMachine v3.

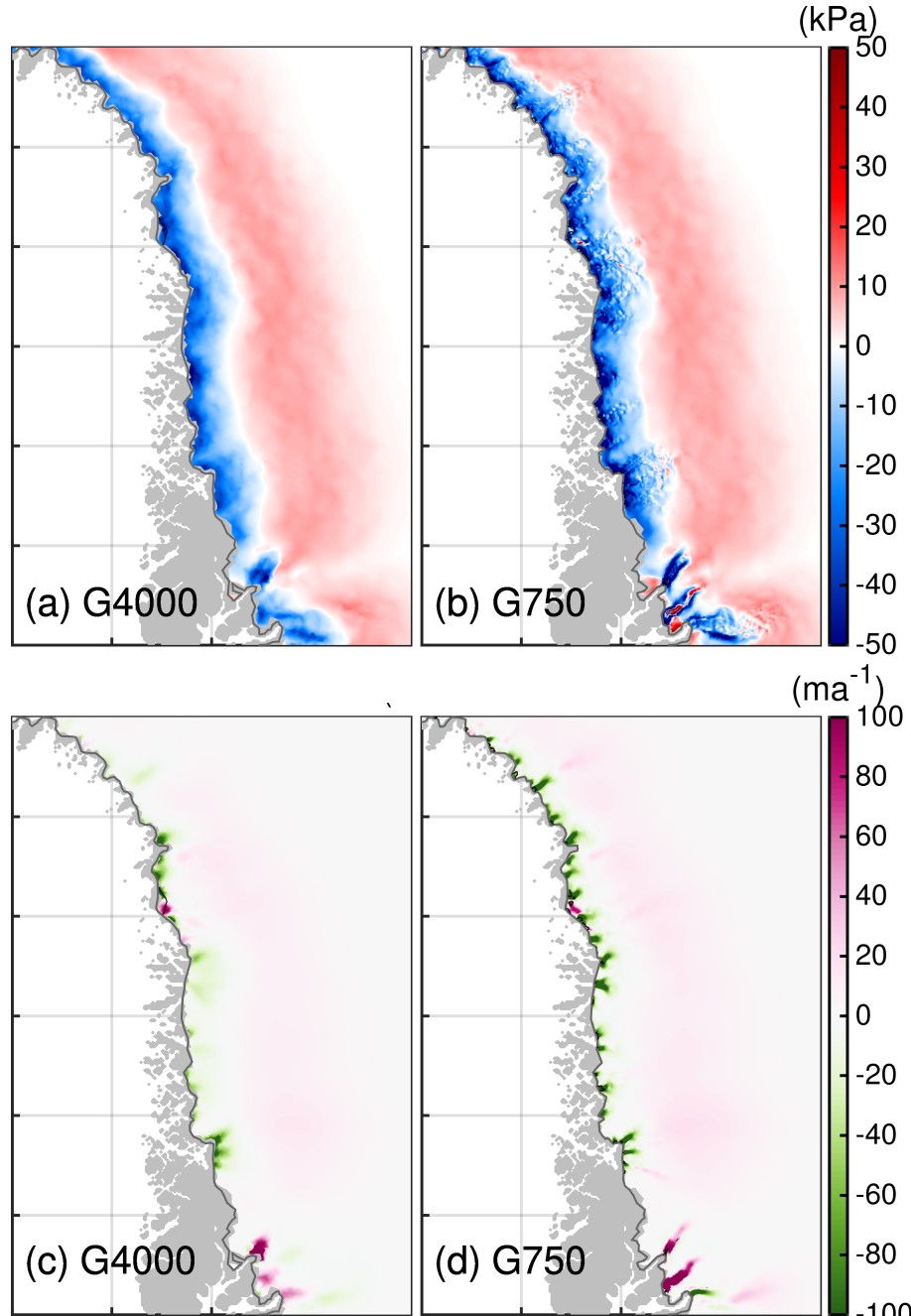

**Figure 13.** (a,b) Difference of driving stress between 2100 and 2015 for the northwestern region. (c,d) Difference of basal velocity between 2100 and 2015 for the northwestern region. Region subsets are shown in Fig. 2. Dark gray line indicates the initial ice extent. Thin black line indicating the grounding line is not visible as it falls together with the calving front. The grey silhouette shows the Greenland land mask from BedMachine v3.

*Code and data availability.* Simulation results on the native grids described in this paper will be made publicly available with a digital object identifier https://doi.org/10.5281/zenodo.3992605. The forcing datasets are available through the ISMIP6 wiki (http://www.climate-cryosphere.org/wiki/index.php?title=ISMIP6_wiki_page, last access: August 21, 2020). The ice flow model ISSM is open source and freely available at https://issm.jpl.nasa.gov/ (last access: August 21, 2020), (Larour et al., 2012). Here ISSM version 4.16 is used.

*Author contributions.* MR conducted the study supported by the other authors. MR set up the ISSM model and ran the experiments. AH analysed the results for the individual glaciers. HG calculated the retreat masks. MR wrote the manuscript together with the other authors.

*Competing interests.* The authors declare that they have no conflict of interest.

*Acknowledgements.* We thank the Climate and Cryosphere (CliC) effort, which provided support for ISMIP6 through sponsoring of workshops, hosting the ISMIP6 website and wiki, and promoted ISMIP6. We acknowledge the World Climate Research Programme, which, through its Working Group on Coupled Modelling, coordinated and promoted CMIP5 and CMIP6. We thank the climate modeling groups for producing and making available their model output, the Earth System Grid Federation (ESGF) for archiving the CMIP data and providing access, the University at Buffalo for ISMIP6 data distribution and upload, and the multiple funding agencies who support CMIP5 and CMIP6 and ESGF. We thank the ISMIP6 steering committee, the ISMIP6 model selection group and ISMIP6 data set preparation group for their continuous engagement in defining ISMIP6 and all their efforts. We thank the Editor Robin Smith and two anonymous reviewers for their helpful suggestions to improve the manuscript. Martin Rückamp acknowledges support of the Helmholtz Climate Initiative REK-LIM (Regional Climate Change). Heiko Goelzer has received funding from the programme of the Netherlands Earth System Science Centre (NESSC), financially supported by the Dutch Ministry of Education, Culture and Science (OCW) under grant no. 024.002.001.

We thank Thomas Kleiner (AWI) and Ralf Greve (ILTS) for continuous and fruitful discussions on simulations. We would like to thank Natalja Rakowsky, Malte Thoma and Bernadette Fritzsch for maintaining excellent computing facilities at AWI and DKRZ. The high-resolution simulations were run on the DKRZ HPC system Mistral under grant ab1073. We also acknowledge the outstanding support of the ISSM team.

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
