# Peer review of "Sensitivity of Greenland ice sheet projections to spatial resolution in higher-order simulations: the AWI contribution to ISMIP6-Greenland using ISSM"

_The Cryosphere, 2019_

## Referee Comment (RC1) · Anonymous Referee #1 · 11 Mar 2020

General comments: The ISMIP6 project is an important effort towards improving projections of the contribution to global sea level rise from the large ice sheets in Greenland and Antarctica. This paper by Rückamp et al. presents result from a particular model from the ISMIP6 project. They show that the projected SLR is sensitive to the spatial resolution in their model, and they show further that the effect depends on the climate forcing, with oceanic and atmospheric forcings showing opposite and non-trivial responses. Investigations of how ice flow modelling techniques and model parameters influence the projected SLR and the uncertainties is highly relevant and very important,

and the main conclusions of this paper are interesting and merits publication.

While the scientific quality of the modelling work is high, the presentation and text are not at the same level. As a result, the purpose of the paper is not clear, the text needs to be edited and self-contained, and the conclusions could be more clearly communicated and supported by the text. The paper appears to be hastily written. Furthermore, the important relation between the resolution of the bedrock topography and the model resolution is not sufficiently discussed. Here are some important issues: 1. The purpose of the paper is mixed and not clear. I recommend that the authors focus on investigating the influence from the spatial resolution and the different effects on oceanic and atmospheric forcings. Please be sure to structure the text to emphasize this purpose. Remove the secondary aim, i.e. to describe how the ISSM-AWI contributed to the IS-MIP6 exercise. This is out of scope here, unless necessary to understand the results of this paper, and should then be part of the methods section. Rewrite the last sections of the introduction to reflect this. The current version seems unclear, particularly the last paragraph, lines 85-91. 2. The introduction starts out being very general and later becomes very specific. Shorten the details of the work by Aschwanden et al., they seem to be too detailed for the introduction. 3. Remove all the additional comments about how the ISSM-AWI model contributed to the ISMIP6 project, e.g. lines 127-126, and several paragraphs in section 2. I don't think that this really supports the conclusions of the paper. 4. Again, comments about how the ISSM-AWI model is set up must be presented in a self-contained way in this paper, so please re-write section 2, and avoid structuring the text as an appendix to the paper by Goelzer et al. Also, remove the first sentence of section 3.3.2, as well as providing details of the ISMIP6 AWI-ISSM6 simulation, which is not reported in this paper. Also, change figure 2 to show results from a run presented here, and not from a G8000 run, which was not reported. 5. The inversion parameters are not discussed in detail, and neither their influence on the simulated velocity field, e.g. how the regularization term smooth out sharp transitions in sliding (in order to avoid oscillations). How does this relate to the spatial resolution, and how sensitive are the results to the regularization parameters? 6. The connection between

the spatial resolution of the model and the ability to resolve bedrock topography should be highlighted more. This work shows that the results converge when the resolution is improved. But is the convergence just because the model resolution approaches the resolution of the bedrock topography map? Would the model results be different if finer resolution bedrock data were available? These are important questions, and they should be discussed in the paper, even if they cannot be fully addressed. In my opinion, the importance of high resolution bedrock data and the relation to the projected SLR is the most important contribution of this paper. Please make this come out more clearly. 7. Throughout the paper: the use of the word "dynamic" is not consistent. Dynamic is used to describe dynamic response, i.e. ice-dynamic changes due to ocean forcing, or dynamic residual, i.e. corrected for front retreat, and sometimes used more generally to describe ice dynamics. Perhaps use the word "discharge" when relevant to avoid confusion. 8. There are numerous errors in the use of English, e.g. proper instead of properly.

A few specific points: - The effective pressure, line 108: is this used generally, i.e. also in interior areas with bedrock below sea level? - Regarding the inversion, please reference the remote sensing velocity product (it is not clear which product is used). - Fig 3b – it is very difficult to see the colored areas. Please modify, e.g. change the black color of grounded ice to white, and remove all black outlines. - Section 4 first paragraph: provide units of the q and Q. - Figures S1+2: confusing caption: difference of a to b, what does that mean – b-a or a-b? please clarify. (difference between a and b is a-b).

---

## Referee Comment (RC2) · Anonymous Referee #2 · 17 Mar 2020

This paper uses the ISSM model and the ISMIP6 protocol in order to investigate the importance of resolving the ice dynamics and the bed topography in Greenland ice sheet simulations. It does so by forcing ISSM using ocean-only retreat, SMB-only perturbations, and the combination of both ocean retreat and SMB perturbations. The paper concludes on the importance of model resolution with respect to the different forcing and how the bed topography resolution can be the ultimate limiting factor on how to choose adequate resolution for model predictions. The findings of this papers are of high quality and highly relevant for improving future projections from ice sheet

models. I would highly support the publication of this paper after revisions suggested thereafter.

General comments:

1. The paper feels that it was written in a hurry, lacks clarity (especially before section 4.2), and could be better organized. There is too much of a focus on ISMIP6 and how the model used here participated in this effort. The paper could simply mention that it extends ISMIP6 contribution by deepening the analysis on the impact of forcing on the model and bed resolution and leave it at that. It is useful to provide a quick summary of the ISMIP6 experimental protocol since it is used as experimental design in the paper but there is no need to add all these references to ISMIP6 throughout the text. I found model descriptions in many different sections while they should be gathered in one place. Similarly, the initialization technics and results should be discussed in a single section with more figures comparing to observations. Terms are misused throughout the text. In the ISMIP6 protocol, what is called "Initial State" in section 4.1 is really the historical run. The control experiment is relative to the state of the initial condition prior to running the historical experiment. The projection control is the one that should be performed starting from the end of the historical run until year 2100 and used in the result analysis section (instead of the control run). It does not appear in the paper and should be added.

2. This paper is very good in discussing and analyzing the results on the model resolution dependence versus the use of grid resolution. It would be of even higher quality if it improved the analysis and the discussion of the dependence with respect to the bed topography versus model resolution (after all, it is major result here).

3. The introduction tries to make a point on the importance of using an adequate resolution for modeling GrIS. However, this is only apparent towards the end of the introduction. It is difficult to follow why it was necessary to read the different paragraphs about initMIP, Greve and Hertzfeld, Aschwanden 2016 and 2019, as no direct conclusions from these finding would lead to the thesis of this study. In particular the paper mentions that the resolution in Greve and Hertzfeld was too coarse to expect any better quantitative results (on top of using SIA). The introduction continues with describing Aschwanden's work that actually did use very high resolution with no real benefit over coarse resolution. At that point, I would expect a discussion of why that is and what this study will do differently.

4. The initialization techniques lack clarity and details. The inversion parameters and their variations with model resolutions are not discussed. After the inversion, is the model run long enough to bring it to a steady state? If so for how long? What dataset (climatology, geothermal heat flux,... ) is used for the initialization? Is there a specific procedure (if any) initiated once the parameters are set? Please add more details on that topic. How are ice shelves constrained during the initialization? A figure highlighting how well the initialized ice sheet matches target observations would be a good addition (the authors mention they want this paper to describe in greater depth how the initialization was performed which could not be done in the ISMIP6 paper by Goelzer et al. 2020).

5. The logic behind not using a sub-grid scheme to simulate fractional retreat in section 3.3.2 does not make sense to me. The argument that it might mimic a higher resolution is not sound as it is something already done for grounding line dynamics. Furthermore, my understanding with how the text is written, is that the grounding line will most likely not coincide with the calving front for grounded marine termini glaciers. The text should be clearer about the description on how the calving front is handled in the model. The grid resolution still plays a role even when a sub-grid scale physics is used in a model. This assumption might play a part in the conclusion and, ideally, the authors should run and present a simulation that suggests otherwise. Also, the model uses an unstructured grid and a straightforward convergence analysis similar to when using a uniform grid is more difficult. Please, revise the argumentation in the text (line 228-230).

6. All the simulations are run with the calving front remaining fixed in space and time besides those with a calving rate mask forcing. This was very confusing as it is not clearly mentioned in the model assumptions in the appropriate section (it only became clear in the result section). The text should make this really clear in the appropriate section of the paper. I do not understand these different treatments and the text does not discuss it. The paper would benefit from more details behind this reasoning, and, also, more discussion on why their conclusions would hold shall this restriction on Rnone experiments be removed. Right now, Rnone experiments do not benefit from the reduced buttressing and from a stronger signal from bed topography adjustments as the other experiments do. The paper mentions this problem but does not discuss it.

7. The discussion about N, tau_b, tau_d, and the sliding velocity in section 4.4 could be extended more. N decreases with the SMB evolution in these experiments. The SMB perturbations lead to a decrease in ice thickness to which N directly depends on, hence a reduction in tau_b. Also, at higher resolution, the marine portion of the glacier shows deeper bed (figure 10) which will result in a lower N, a lower basal friction and an increase in sliding velocity in order to balance the driving stress. Gagliardini et al. 2007, and Leguy et al. 2014 study these relationships and they can be used as references for the discussions. Also, this discussion item should be tied in with the discussion of the importance of the bed resolution especially when using effective pressure dependent basal friction laws. Oddly enough, these points are being mentioned in the conclusion but not before, why?

Specific comments:

Page 1: Line 16: remove the character "N". Line 14-16: "A major response ..." By invoking the sliding mechanism using effective pressure you are inherently talking about the dependency with respect to the basal sliding law used in the model. I would simply live it as that in the abstract as there is no further modeling details given at that point.

Page 2: line 34: add the citation of Nowicki et al. 2020. line 44: Please rephrase. line

46: replace "affect" by "affects". line 54: replace "well" with "will". last paragraph (line 50-54): be careful with the first sentence here as Fig. 1 clearly shows (with ISSM) that a resolution of 0.5 km was necessary to see a drop in SL contribution and no other models (in this figure) submitted results at that resolution. Also, please clarify that the importance of resolving the ice margins in the initMIP simulations is because they are subjected to the strongest SMB anomalies and SMB anomaly transitions compared to the interior of the ice sheet.

Page 3: Line 59: please spell out SIA as it is the first time it is employed in the text. Line 70: "however, the SMB..." please clarify what it means and why it matters here. Last sentence: Why is this information of importance? Are you trying to make a point that their choice of Stokes approximation is a limiting factor? As stated, I would simply remove it.

Page 4: Line 74: "which is . . . (Church et al., 2013)" this comment feels out of place in what you are trying to say here. I would remove it. Line 77: "The adequate resolution..." please add citation(s) to support this claim. Also, as stated it is quite confusing because increasing the resolution is a good thing (up to a certain point) regardless of the Stokes approximation. The resolution dependency is typically greater with sub-grid scale physical mechanism such as grounding line tracking, . . . or when needed to better resolve bed topography. Line 78: "higher-order approximation is providing..." please add citation(s) supporting your claim (for similar reason as previous remark). Line 82: "to this task" please clarify what you mean here. Line 82: "Therefore, the main..." I would suggest beginning a new paragraph with this sentence adding directly what will be the major difference compared to what Aschwanden did (which is very similar). Line 85: "Blatter-Pattyn-type" Is it different than BP? If so, how does it differ? Otherwise remove "type". Please, add a reference to BP here as it is the first time you mention it and you can remove the one on line 99. Line 87: "For comparison..." What is the relevance of this information here. Line 90: "A secondary aim . . ." This sentence is confusing here as it sounds like the aim of this paper is to

redo the ISMIP6 exercise. Line 91: "which could be valuable..." not necessary there. Line 91: the footnote on the word "audience" This footnote is confusing. Are there any differences between ISSM and AWI-ISSM? If so the text should highlight these differences to improve clarity. For instance, which release of ISSM did AWI branch from? Was there major development(s) made since then and if so add a reference. Line 99: Blatter-Pattyn is a very expensive model to run. Please clarify what you mean by "balancing computational cost", are you referring in comparison to full Stokes? Line 102: please add citations for the characteristics of the model (Glen's flow law, temperature dependent rate factor ...).

Page 5: Line 104: add "," after "base". Line 104: please add a citation for this form of sliding law. Also, please clarify your choice of sliding law. This formulation is typically avoided as it can grow unboundedly (schoof 2005). Also, it would be good to provide a map of the $k^2$ friction coefficient. Line 109: "At lateral..." The sentence is confusing, please reword. Line 111: Please provide a citation or link for EPSG:3413 grid. Line 122: Please indicate if the grid is fixed throughout the simulation or evolving. Line 124-125: typically, modelers think of high resolution being the smallest mesh size used in a model and the coarse (low) resolution being the biggest one. It is less confusing for RESmin to be the coarsest resolution and RESmax to be the highest. Line 127: "Additionally, we ..." This information is out of place here and should be omitted.

Page 6: Line 136: The sentence here contradicts the title of section 3. Maybe rename section 3 as "Forcing experiments" or something similar, and simply state that you are following the ISMIP6 experimental design. Line 138: I suggest writing "Slater et al. (2019a, b)" similarly to what you did on page 9 line 222. Line 139-142: Why is it necessary to mention initMIP here?

Page 7: Line 152: there is also a projection control experiment that starts at the end of the historical run. Have you run it? Line 155: "The ensemble..." I believe it refers to the ensemble from ISMIP6? If so this sentence does not add any value to the paragraph. Line 159: Please briefly recall how low, median, and high oceanic forcing

were defined. Paragraph 3: "Conducted projection . . ." This paragraph is out of place and should be combined somehow with section 3.3.2. The definitions of the runs (which are highlighted in Table2) could be given at the beginning of the result section.

Page 8: Line 191: "That means. . ." This sentence is confusing. Do you mean that grounded and floating ice cells are not allowed to retreat? If so it restricts the purpose of the historical run. Please clarify.

Page 9: Line 224: "The imposed . . ." Is this sentence supposed to explain how the prescribed calving front retreat was obtained? If so, say so.

Page 10: Line 228: "This enables . . ." See the general comments. Additionally, this statement is ambiguous because you are using an unstructured grid. While you can compare the results from the simulation using different grids, you cannot claim your comparison to be consistent to grid resolution. Please rephrase. Line 237: The title of section 4.1 reads "Initial state". This title Is confusing. Typically, the initial state is the one obtained at the end of the inversion procedure and the one used as initial condition for the historical and control runs. Please rephrase. Section 4.1: this section contains information that should be stated in section 3.1 such as the restriction of the calving front during the inversion procedure . . .

Page 11: Line 261: "Similar as . . ." There is no need to repeat this sentence here since the MSD metric is used again. Line 268: "As the ice . . ." Please discuss further the reason of keeping the calving front fixed throughout the historical run. Line 274: "with the control" The projection runs should be corrected with a projection control run instead which is not discussed in this paper. Line 275: "in the absence of additional forcing" This defines the control run. It is an unnecessary repetition. Line 276: ". . . as a prediction of actual behavior . . ." This is out of place because the text is talking about the control and have not induced any forcing yet. Please rephrase.

Page 12: Line 279: replace "simulation" with "simulations". Line 282: replace "with" with "to". Line 282: "(see above)" Please refer to a section for clarity (unless you are

referring to the mass gain numbers?).

Page 14: Line 304: replace "compared the total" with "compared to the total". Line 307: remove the repetition of "the". Line 311: replace "RCP8.5-Rnone" with "RCP8.5-Rnone and RCP8.5-Rlow"? Line 316: replace "lesser than" with "less than".

Page 15: Line 347: reword "early in the century an increase" with "an increase early in the century".

Page 16: Line 357: replace "worth to mention" with "worth mentioning". Line 359: "remains fixed in time …" See my general comment. Line 366: replace "reduce" with "reduces,". Line 367: replace "not obvious" with "non obvious". Line 368: remove "come into play". Line 369: "The general picture …" Please rephrase.

Page 17: Line 378: "To study … grid size" Please rephrase. Line 392: replace "together an increase" with "together causing an increase"? Line 392: replace "thinning an acceleration" with "thinning and acceleration"? Line 392: "The transient …" Please rephrase the end of this sentence.

Page 18: Line 400: replace "nasal" with "basal". Line 405: add "we" before "find". Line 416: please rephrase end of sentence. Line 423: replace "it is worth to investigate this influence isolated" with "it will be worth investigating this influence only"?

Page 19: Line 428: replace "in numerous cases" with ", in numerous cases,"? Line 431: replace "assessing the importance of it" with "assessing its importance"? Line 434: remove "thus"?

Tables: Table 1: Is the computational time listed here for all the experiments or simply for the 86-year run after the historical run?

Figures: In the relevant figures, please add a black contour for the grounding line.

Figure 2: replace "G8000" with "G4000". The small ice cap above 79N should not be present for consistency with the text and the other figures in the paper. Figure 5: it

should really be Figure 6 since its reference appear after figure 6 in the text. Figure 6: it should be relabeled Figure 5 (see figure 5 comment above). Figure 9: the subfigure labels b and c are misplaced. What are the units for Year? (I have never seen CE before as a unit). Figure 10: the x-axis is labeled "distance". What is it relative to? Please add this reference to the figure. Also, please try to increase the font size of the labels as they are difficult to read on printed paper.

---

## Author Comment (AC1) · 18 May 2020

The comment was uploaded in the form of a supplement:
https://www.the-cryosphere-discuss.net/tc-2019-329/tc-2019-329-AC1-
supplement.pdf

---

## Author Comment (AC2) · 18 May 2020

The comment was uploaded in the form of a supplement:
https://www.the-cryosphere-discuss.net/tc-2019-329/tc-2019-329-AC2-supplement.pdf

---

## Author Response (AR1)

We thank both reviewers for their supportive and thorough review, and acknowledge all of the points made. Most of the comments concern the presentation and the discussion of the results. We agree on that and aim to better present the results in a revised version of the manuscript. We highly appreciate the suggestions and follow most of the comments to provide a better and self-contained manuscript so that the purpose of the research and results become much clearer.

Due to the comments, the revised version of the manuscript will have the following major changes:

- The second part of the Introduction is rewritten. A major consequence is that the main focus of the paper is now on the grid dependency. We dropped the (secondary) aim to present the AWI-ISSM model in detail. So, we still present more details of AWI-ISSM compared to Goelzer et al (2020), but we put more weight on the grid-dependence.
- The *Discussion* is improved. We identified paragraphs in the *Results* section, which should belong to the *Discussion* (this might have also confused the reviewers - apologies). We moved them accordingly. The original paragraphs were found on page 14, line 320-323; page 16, line 353-364 and on page 17, line 378-383 in the first version of the manuscript, see point-to-point answers below.
- We have added a new subsection focussing on the comparability of the experiments.
- As requested by the reviewers, we re-structured the description of the ISM used and description of the experiments. With that, we hope to gather all information in the appropriate sections. The structure now looks as:

   2. Methods and experiments
      2.1 Ice flow model ISSM
      2.2 Overview of experiments
      2.3 Initialization experiment
      2.4 Historical experiment
      2.5 Future forcing experiments
         2.5.1 Atmospheric forcing
         2.5.2 Oceanic forcing
      2.6 Comparability of experiments **(new)**

We reply to each point made (reviewer text = black text) below using blue text. The changes to the manuscript will be found at the end of the responses to the reviewer comments.

==== Reviewer 1 =====

General comments: The ISMIP6 project is an important effort towards improving projections of the contribution to global sea level rise from the large ice sheets in Greenland and Antarctica. This paper by Rückamp et al. presents result from a particular model from the ISMIP6 project. They show that the projected SLR is sensitive to the spatial resolution in their model, and they show further that the effect depends on the climate forcing, with oceanic and atmospheric forcings showing opposite and non-trivial responses. Investigations of how ice flow modelling techniques and model parameters influence the projected SLR and

the uncertainties is highly relevant and very important, and the main conclusions of this paper are interesting and merits publication. While the scientific quality of the modelling work is high, the presentation and text are not at the same level. As a result, the purpose of the paper is not clear, the text needs to be edited and self-contained, and the conclusions could be more clearly communicated and supported by the text. The paper appears to be hastily written. Furthermore, the important relation between the resolution of the bedrock topography and the model resolution is not sufficiently discussed.

We would like to thank the reviewer for his/her generally positive evaluation. The comments are very helpful to clean the previous version of the manuscript and provide a better and concise story.

Here are some important issues:

1. The purpose of the paper is mixed and not clear. I recommend that the authors focus on investigating the influence from the spatial resolution and the different effects on oceanic and atmospheric forcings. Please be sure to structure the text to emphasize this purpose.
We have restructured the text , i.e. experiment description and aim of paper, as outlined above. We believe with restructuring the text, that the purpose of the paper to investigate the influence from the spatial resolution on future forcing experiments is now more highlighted.

Remove the secondary aim, i.e. to describe how the ISSM-AWI contributed to the ISMIP6 exercise. This is out of scope here, unless necessary to understand the results of this paper, and should then be part of the methods section.
You are right, we deleted all unnecessary information e.g. how AWI-ISSM contributed to ISMIP6. In doing so, we now emphasize the grid-dependency as major aim.

Rewrite the last sections of the introduction to reflect this. The current version seems unclear, particularly the last paragraph, lines 85-91.
To emphasize the focus of the paper, we modified the last paragraph of the *Introduction* (Lines 73-81 in the new version of the manuscript).

2. The introduction starts out being very general and later becomes very specific. Shorten the details of the work by Aschwanden et al., they seem to be too detailed for the introduction.
We agree that there are too many details in the *Introduction*. We have shortened and reorganized the material in the second part of the *Introduction*.

3. Remove all the additional comments about how the ISSM-AWI model contributed to the ISMIP6 project, e.g. lines 127-126, and several paragraphs in section 2. I don't think that this really supports the conclusions of the paper.
We have reorganized the material and focus on the grid-dependency.

4. Again, comments about how the ISSM-AWI model is set up must be presented in a self-contained way in this paper, so please rewrite section 2, and avoid structuring the text as an appendix to the paper by Goelzer et al. Also, remove the first sentence of section

3.3.2, as well as providing details of the ISMIP6 AWI-ISSM6 simulation, which is not reported in this paper. Also, change figure 2 to show results from a run presented here, and not from a G8000 run, which was not reported.

We completely rearranged the material to present a better and self-contained story.
Not exactly sure what you mean with AWI-ISSM6, as it does not appear in the manuscript. We think the comment refers to AWI-ISSM1? If so, the reference to AWI-ISSM1 is deleted.
"G8000" was a typo and now changed to "G4000".

5. The inversion parameters are not discussed in detail, and neither their influence on the simulated velocity field, e.g. how the regularization term smooth out sharp transitions in sliding (in order to avoid oscillations). How does this relate to the spatial resolution, and how sensitive are the results to the regularization parameters?

We have not explored in depth how the parameters will change across the different spatial resolutions. To find the optimal inversion parameters, we need to run an L-curve analysis for each grid. For a high-resolution setup this was already done by Seroussi et al. (2013) and we adopt these parameters (see page 8, line 181). To run an L-curve analysis for the coarser resolutions is beyond the scope of this study. In fact, we wanted to keep all parameters similar and argue that the simulations should therefore be comparable (for the same reason, we do not induce a SMB correction to minimize the drift in the control run). We added a section "Comparability of experiments" to highlight our approach (Lines 232-244 in the new version of the manuscript). Additionally, we refer in the *Discussion* to this issue (Lines 469-476 in the new version of the manuscript).

6. The connection between the spatial resolution of the model and the ability to resolve bedrock topography should be highlighted more. This work shows that the results converge when the resolution is improved. But is the convergence just because the model resolution approaches the resolution of the bedrock topography map? Would the model results be different if finer resolution bedrock data were available? These are important questions, and they should be discussed in the paper, even if they cannot be fully addressed. In my opinion, the importance of high resolution bedrock data and the relation to the projected SLR is the most important contribution of this paper. Please make this come out more clearly.

We extended the discussion on page 18, line 425 between the spatial resolution of the model and the ability to resolve bedrock topography. Also we moved the paragraph from page 17, line 378-383 to the discussion. This paragraph particularly addresses one of your questions. We discussed a setup that adopted a high-resolution grid (G1000), but uses input data (e.q. bedrock) from a coarse resolution (G4000). This scenario shows a sea-level contribution that is closer to the G4000 simulation. This setup highlights that the response is highly dependent on how the bedrock is resolved. We believe by extending and restructuring the discussion, we have made this much clearer (Lines 392-432 and 438-450 in the new version of the manuscript).

7. Throughout the paper: the use of the word "dynamic" is not consistent. Dynamic is used to describe dynamic response, i.e. ice-dynamic changes due to ocean forcing, or dynamic residual, i.e. corrected for front retreat, and sometimes used more generally to describe ice dynamics. Perhaps use the word "discharge" when relevant to avoid confusion.

You are right, at some instances the use of "dynamic" was not consistent. We have corrected where necessary.

8. There are numerous errors in the use of English, e.g. proper instead of properly.
We have carefully read the manuscript and corrected the use of English where necessary.

A few specific points:
- The effective pressure, line 108: is this used generally, i.e. also in interior areas with bedrock below sea level?
Yes, we do not make a separation if areas are in contact with the ocean or not. The parametrization accounts for full water-pressure support from the ocean wherever the ice sheet base is below sea-level, even far into its interior where such a drainage system may not exist. We acknowledge, that this is a strong simplification of the model but an additional hydrological model would be needed to realistically simulate the effective basal pressure.
We clarified it in the new version and add a paragraph in the discussion (Lines 461-468 in the new version of the manuscript).

- Regarding the inversion, please reference the remote sensing velocity product (it is not clear which product is used).
Indeed, there was a citing error. We changed Joughin et al. (2010) to Joughin et al. (2018) as requested by NSIDC (https://nsidc.org/data/NSIDC-0670/versions/1) for citing *MEaSUREs Multi-year Greenland Ice Sheet Velocity Mosaic, Version 1.*

- Fig 3b – it is very difficult to see the colored areas. Please modify, e.g. change the black color of grounded ice to white, and remove all black outlines.
We have modified the colors. Instead of the showing the mask for the whole GrIS, we show only a subset.

- Section 4 first paragraph: provide units of the q and Q.
Done

- Figures S1+2: confusing caption: difference of a to b, what does that mean – b-a or a-b? please clarify. (difference between a and b is a-b).
Done.

======= Reviewer 2 =======

This paper uses the ISSM model and the ISMIP6 protocol in order to investigate the importance of resolving the ice dynamics and the bed topography in Greenland ice sheet simulations. It does so by forcing ISSM using ocean-only retreat, SMB-only perturbations, and the combination of both ocean retreat and SMB perturbations. The paper concludes on the importance of model resolution with respect to the different forcing and how the bed topography resolution can be the ultimate limiting factor on how to choose adequate resolution for model predictions. The findings of this papers are of high quality and highly relevant for improving future projections from ice sheet models. I would highly support the publication of this paper after revisions suggested thereafter.

Thank you very much for the positive evaluation. Your detailed comments below are highly appreciated to improve the presentation of the manuscript. In most instances we followed your suggestions. A few suggestions arised from a misleading presentation in the first version of the manuscript, therefore we have not adopted them but clarified the presentation. The suggestion in comment 7, we have realized to that extent that we have polished the *Discussion.*

General comments:

1. The paper feels that it was written in a hurry, lacks clarity (especially before section 4.2), and could be better organized. There is too much of a focus on ISMIP6 and how the model used here participated in this effort. The paper could simply mention that it extends ISMIP6 contribution by deepening the analysis on the impact of forcing on the model and bed resolution and leave it at that. It is useful to provide a quick summary of the ISMIP6 experimental protocol since it is used as experimental design in the paper but there is no need to add all these references to ISMIP6 throughout the text. I found model descriptions in many different sections while they should be gathered in one place.
In the updated manuscript, we shortened the description of the ISMIP6 protocol and tried to group this type of information. The restructuring was also requested by Reviewer 1 (see his/her comments 1,3 and 4).

Similarly, the initialization technics and results should be discussed in a single section with more figures comparing to observations.
Reviewer 1 recommended dropping the AWI-ISSM details as one of the aims of the paper to make the grid sensitivity much clearer. As we followed his/her suggestion we decided that we will not present the performance of the initialization in detail. However, we updated the histogram plots in Fig. 4 by showing the RMSE values of the initialization. The text is updated accordingly. Additionally we added new figures to the supplement (Figs. S1, S2 and S3).

Terms are misused throughout the text.
We have carefully read the manuscript and corrected terms where necessary.

In the ISMIP6 protocol, what is called "Initial State" in section 4.1 is really the historical run.
Indeed, we renamed section 4.1 "Initial state" to "Historical experiment".

The control experiment is relative to the state of the initial condition prior to running the historical experiment. The projection control is the one that should be performed starting from the end of the historical run until year 2100 and used in the result analysis section (instead of the control run). It does not appear in the paper and should be added.
In the revised version of the manuscript we define additionally the ctrl_proj experiment, similar as in Goelzer et al. (2020). When correcting the projections with ctrl_proj the magnitude of the SL contributions in 2100 of each experiment changes, but not the general behaviour (illustrated in the Figures above; equivalent to Figs. 5 and 7 in the manuscript but corrected with ctrl_proj). We added a table that lists all SL contributions from each

experiment corrected with both ctrl and ctrl_proj. However, in the revised version the results of sea-level contribution (Figs. 5 and 7) are still corrected with the ctrl run.

2. This paper is very good in discussing and analyzing the results on the model resolution dependence versus the use of grid resolution. It would be of even higher quality if it improved the analysis and the discussion of the dependence with respect to the bed topography versus model resolution (after all, it is major result here).
We have done this and described it in our answer to comment 6 of Reviewer 1.

3. The introduction tries to make a point on the importance of using an adequate resolution for modeling GrIS. However, this is only apparent towards the end of the introduction. It is difficult to follow why it was necessary to read the different paragraphs about initMIP, Greve and Hertzfeld, Aschwanden 2016 and 2019, as no direct conclusions from these finding would lead to the thesis of this study. In particular the paper mentions that the resolution in Greve and Hertzfeld was too coarse to expect any better quantitative results (on top of using SIA). The introduction continues with describing Aschwanden's work that actually did use very high resolution with no real benefit over coarse resolution. At that point, I would expect a discussion of why that is and what this study will do differently.
The intention of describing iniMIP, Greve & Hertzfeld 2013, and Aschwanden et al. (2016,2019) was to outline the importance of the horizontal grid-resolution and outlining wat is known about the grid-dependency on GrIS projections. We agree that we present too many details which are not necessary at this stage. We have now shortened the description and mention very briefly, that (1) the observed grid-dependency in iniMIP must be treated with some caution due to the methodological approach and that (2) in Greve & Hertzfeld (2013) and Aschwanden et al. (2019) no clear conclusion how the resolution affects the mass loss was found. From here we deduce one research question, that a separation of both forcings (atmosphere and ocean) must be considered to investigate the different responses (Lines 56-61 in the new version of the manuscript).

4. The initialization techniques lack clarity and details. The inversion parameters and their variations with model resolutions are not discussed.
We improved this considerably. Please see answer to comment 5 of Reviewer 1.

After the inversion, is the model run long enough to bring it to a steady state? If so for how long?

After the inversion we directly started the historical run. We don't perform any further relaxation run to bring the model to a steady-state. It is always a compromise between matching the observed geometry and being closer to a steady-state. Here, we put more weight on having the initial geometry closer to the observed geometry. This decision is-line with our approach having all parameters, parameterizations and inputs for all grids as similar as possible for a better comparison between the resolutions. As the model is not in steady-state at the initial state, we expect a model drift in the transient runs; which would not the case for models that do a relaxation towards a steady-state after the inversion. To demonstrate the grid-dependent drift, we show the spatial patterns in Fig. (6).

We added: *"All transient simulations start from the initial state, that means, we do not perform a subsequent relaxation run to bring the model to a steady-state. We put more weight on having the geometry closer to the observed geometry for a better comparison between the resolutions."*

What dataset (climatology, geothermal heat flux, ... ) is used for the initialization?

Geothermal flux data set is mentioned on page8 line 175. However, we would like to clarify that there is no thermo-mechanical coupling in the transient runs. Furthermore, no climatology/geothermal flux is used in our initialization approach. With the temperature field recovered from Rückamp et al. (2019) we just initialize the viscosity with a realistic temperature distribution rather than choosing e.g. a constant temperature (which is often done in inversion approaches). If climatology refers to which SMB product is used, that is outlined very detailed in section 3.3.1.

Is there a specific procedure (if any) initiated once the parameters are set? Please add more details on that topic. How are ice shelves constrained during the initialization? A figure highlighting how well the initialized ice sheet matches target observations would be a good addition (the authors mention they want this paper to describe in greater depth how the initialization was performed which could not be done in the ISMIP6 paper by Goelzer et al. 2020).

Our initialization procedure is based on data assimilation. That means we leverage BedMachine as geometric input and MEASURE velocities as target for the inversion method. Instead of showing the performance of the initialization (v_simulated vs. v_observed and thickness_simulated vs thickness_observed), we present the quality at the end of the historical run (Fig. 4). As the historical run causes some geometrical changes and velocity responses (as response to the imbalance of ice sheet which is not in equilibrium with the applied SMB and ice flux divergence), the misfits are a bit larger than directly after the initialization.

As we focused now more on the grid-dependency rather than on model description (as suggested by Reviewer 1), we tried to keep the description of the initialization procedure as briefly as possible. However, we slightly rewrote the initialization procedure to clarify certain aspects. Due to dropping the focus of the paper to present the AWI-ISSM details, we decided that we will not present the performance of the initialization in detail. However, we updated the histogram plots in Fig. 4 by showing the RMSE values of the initialization. Text is updated accordingly. Additionally we added new figures to the supplement (Figs. S1, S2 and S3).

5. The logic behind not using a sub-grid scheme to simulate fractional retreat in section 3.3.2 does not make sense to me. The argument that it might mimic a higher resolution is not sound as it is something already done for grounding line dynamics.

Our intention was to highlight that the coarser resolution models requesting a sub-grid scheme for the retreat according to the ISMIP6 protocol. The fractional retreat than mimics a higher resolution what we do not want here because the coarser models must rely on the parameters, parameterizations, inputs etc. that were tuned for the highest resolution (G750). This is our comparison approach now outlined in the new section "Comparability of experiments". The sub-grid scheme for the grounding line is enabled for the G750 run, therefore we also enable it for the coarser resolutions. For the G750 model, the ISMIP6 protocol do not request a sub-grid scheme for retreat, therefore no sub-grid retreat is enabled for the coarser models. We have rephrased the sentences for clarification.

Furthermore, my understanding with how the text is written, is that the grounding line will most likely not coincide with the calving front for grounded marine termini glaciers.

In most instances the calving front coincide with grounding line. We have now clarified this in the section "Ice flow model ISSM": "*However, at most locations the grounding line coincides with the calving front. Except for the floating tongue glaciers Petermann, Ryder and 79° North, the sub-grid schemes at the grounding line is not applied. The treatment of the calving front evolution depends on the experimental setup and is explained in Sect. 2.3 and 2.4.2.*"

The text should be clearer about the description on how the calving front is handled in the model.

In the new version of the manuscript, the sections "2.4 Historical and control experiments" and "2.5.2 Oceanic forcing" now clearly state how the calving front is handled. For the historical, ctrl and ctrl_proj the calving front remains fixed. In the projection experiments the calving front is forced with the retreat masks.

The grid resolution still plays a role even when a sub-grid scale physics is used in a model. This assumption might play a part in the conclusion and, ideally, the authors should run and present a simulation that suggests otherwise.

As our statement about using sub-grid scale physics for the GL but not for the retreat was misleading, we think this comment is obsolete. However, we run a few simulations where we vary the GL treatments which are currently available in ISSM. These experiments reveal that the absolute magnitude of mass loss changes very slightly but the general trend is not altered.

Also, the model uses an unstructured grid and a straightforward convergence analysis similar to when using a uniform grid is more difficult. Please, revise the argumentation in the text (line 228-230).

We added a new section that addresses the "Comparability of the simulations" (see answer to general comment 5 of Reviewer 1). With restructuring the description of the model and the experiment we hope to clarify how the calving front is treated.

6. All the simulations are run with the calving front remaining fixed in space and time besides those with a calving rate mask forcing. This was very confusing as it is not clearly mentioned

in the model assumptions in the appropriate section (it only became clear in the result section). The text should make this really clear in the appropriate section of the paper.

To be clear, the ctrl, ctrl_proj, the historical and the RCP8.5-Rnone scenarios are run with a fixed calving front. The other projection runs (RCP8.5-Rlow/med/high and OO-Rlow/med) received the retreat parametrization for marine terminating outlet glaciers. We hope with restructuring the appropriate sections this will become clearer now. To keep the atmospheric and oceanic forcing fixed in the control run was a request by the ISMIP6 protocol. In our case the ice sheet mask remains fixed in time and space (i.e. no advance or retreat of the calving front).

For the control experiments we added the following sentence: *"In both control experiments (ctrl and ctrl_proj) the SMB and ice sheet mask remains unchanged to the reference year according to the ISMIP6 protocol."*

For simplicity, we choose the oceanic forcing fixed in the historical scenario. However, we added: *"This is a crude approach but representing the historical mass loss accurately was not a strong priority for our experimental setup."*

I do not understand these different treatments and the text does not discuss it. The paper would benefit from more details behind this reasoning, and, also, more discussion on why their conclusions would hold shall this restriction on Rnone experiments be removed. Right now, Rnone experiments do not benefit from the reduced buttressing and from a stronger signal from bed topography adjustments as the other experiments do. The paper mentions this problem but does not discuss it.

Thanks, that is indeed a very good point that is now better discussed. As now mentioned in the Introduction, we choose the different forcings (retreat activated and not activated) to assess the effect of the oceanic forcing on the different grids separately. We extended the discussion (see lines 416-432).

7. The discussion about N, tau_b, tau_d, and the sliding velocity in section 4.4 could be extended more. N decreases with the SMB evolution in these experiments. The SMB perturbations lead to a decrease in ice thickness to which N directly depends on, hence a reduction in tau_b. Also, at higher resolution, the marine portion of the glacier shows deeper bed (figure 10) which will result in a lower N, a lower basal friction and an increase in sliding velocity in order to balance the driving stress. Gagliardini et al. 2007, and Leguy et al. 2014 study these relationships and they can be used as references for the discussions. Also, this discussion item should be tied in with the discussion of the importance of the bed resolution especially when using effective pressure dependent basal friction laws. Oddly enough, these points are being mentioned in the conclusion but not before, why?

We would like to refrain from deepening the analysis to the balance of stresses, as it requires further an analysis of e.g. longitudinal and lateral drag changes as the driving stress is not fully balanced by basal drag in fast flowing areas. Although such an analysis is very important to gain further understanding of the underlying processes it is beyond the scope of this study. Our intention with presenting tau_b and tau_d was to illustrate ongoing changes over the course of the experiments. In doing so, these insights help to understand the driving causes and mechanisms between the employed grids. However, we reorganized the discussion (e.g. the points from the conclusion have now made it into the discussion) and we add new material (Figs. 3, 12 and 13 in the new version of the manuscript) to support our

conclusions. The rephrased section are found in lines 392-407 in the new version of the manuscript. Additionally we add a brief discussion about the friction law used (lines 461-468).

Unfortunately, we placed a slightly wrong statement in the conclusion ("*In general, a reduction of N is overcompensated by a reduction in tau_b, leading to an increase in sliding speed*", Line 452/453). We detected a non-trivially response: in RCP8.5-Rnone most of the outlet glaciers experience a slow-down along with a decrease in tau_b. For scenarios with considered retreat, we observe a widespread glacier acceleration along with a increase auf tau_b. The latter means, in the coevolving fields N and v_b, the increase of v_b overcompensates the decrease of N (such a behaviour would probably not apparent using a bounded friction law). Please note, that we have adjusted the colormap showing the changes in tau_d and tau_b (Figs. S20 and S21). Reddish (+) shows now an increase and blueish decrease (-).

The connection between N and the bed is not as simple as outlined in your comment. Of, course the marine portions in the fine resolution are much deeper as in the coarser resolution. But the ice thickness is also better resolved at higher resolution. Therefore, the deeper troughs/fjords alone do not directly cause a lower N (see figures below showing $p\_ice=rho\_i*g*thickness$ (a,b) and N (c,d) for G4000 and G750 in a selected region).

[Figure]

Specific comments:

Page 1: Line 16: remove the character "N".
Done

Line 14-16: "A major response ... " By invoking the sliding mechanism using effective pressure you are inherently talking about the dependency with respect to the basal sliding law used in the model. I would simply live it as that in the abstract as there is no further modeling details given at that point.
We agree and dropped "(despite no climate-induced hydrological feedback is invoked)"

Page 2: line 34: add the citation of Nowicki et al. 2020.
We added this reference.

Line 44: Please rephrase.

Done.

Line 46: replace "affect" by "affects".
Done

line 54: replace "well" with "will".
Done.

last paragraph (line 50-54): be careful with the first sentence here as Fig. 1 clearly shows (with ISSM) that a resolution of 0.5 km was necessary to see a drop in SL contribution and no other models (in this figure) submitted results at that resolution. Also, please clarify that the importance of resolving the ice margins in the initMIP simulations is because they are subjected to the strongest SMB anomalies and SMB anomaly transitions compared to the interior of the ice sheet.
We have rewritten the paragraph to:
*"Interestingly, the estimated sea-level contributions show a dependence on grid resolution (Fig. 1). ISM versions with multiple grid resolutions demonstrate that coarser grid resolutions tend to produce a slightly larger mass loss. However, this effect is partly due to the methodological approach by considering a SMB anomaly that is based on the present-day observed SMB. Therefore, ISMs with initial areas larger than observed are subject to more and stronger melting and sharper transitions in SMB. Therefore, coarse resolution models not rendering the present-day ice margin perfectly will likely overestimate ablation."*

Page 3: Line 59: please spell out SIA as it is the first time it is employed in the text.
Done. "SIA" does not appear any longer in the text.

Line 70: "however, the SMB ... " please clarify what it means and why it matters here. Last sentence: Why is this information of importance? Are you trying to make a point that their choice of Stokes approximation is a limiting factor? As stated, I would simply remove it.
As suggested, the sentence is deleted.

Page 4: Line 74: "which is ... (Church et al., 2013)" this comment feels out of place in what you are trying to say here. I would remove it.
Done. Sentence is removed.

Line 77: "The adequate resolution ... " please add citation(s) to support this claim. Also, as stated it is quite confusing because increasing the resolution is a good thing (up to a certain point) regardless of the Stokes approximation. The resolution dependency is typically greater with sub-grid scale physical mechanism such as grounding line tracking, ... or when needed to better resolve bed topography.
We have rewritten the sentences to:
*"High-resolution models, in turn, require a larger amount of computational resources. Additionally, when increasing the resolution, simple approximations to the momentum balance do not provide an accurate solution (Pattyn et al., 2008). This limitation takes place particularly at the ice sheet margin and at outlet glaciers where all terms in the force balance become equally important (e.g. Pattyn and Durand, 2013). Due to the intensive*

*computational resources needed to solve the full-Stokes equation, higher-order approximations provide a good compromise to balance model accuracy and computational costs on centennial time scales."*

Line 78: "higher-order approximation is providing ... " please add citation(s) supporting your claim (for similar reason as previous remark).
See answer to Line 77

Line 82: "to this task" please clarify what you mean here.
We replaced "task" with "grid resolution".

Line 82: "Therefore, the main ... " I would suggest beginning a new paragraph with this sentence adding directly what will be the major difference compared to what Aschwanden did (which is very similar).
We do not exactly rewrite the lines as suggested but we mentioned now earlier in the Introduction (as a consequence to Aschwanden's work):
*"... A separation of both responses in future projections experiments would shed some light, how these two main contributors from the GrIS to sea-level are affected by the horizontal resolution…."*
In response to the reviewers comment we write here: *"... Beside running the full scenarios (i.e. both oceanic and atmospheric forcing considered), we aim to explore the grid resolution dependence on atmospheric and ocean forcing separately. ..."*

Line 85: "Blatter-Pattyn-type" Is it different than BP? If so, how does it differ? Otherwise remove "type". Please, add a reference to BP here as it is the first time you mention it and you can remove the one on line 99.
Thanks. We changed it accordingly.

Line 87: "For comparison ... " What is the relevance of this information here.
We dropped the sentence.

Line 90: "A secondary aim ... " This sentence is confusing here as it sounds like the aim of this paper is to redo the ISMIP6 exercise.
The sentence is dropped. As suggested by Reviewer 1, the main focus is now on the grid-dependence sensitivity.

Line 91: "which could be valuable ... " not necessary there.
We dropped this sentence.

Line 91: the footnote on the word "audience" This footnote is confusing. Are there any differences between ISSM and AWI-ISSM? If so the text should highlight these differences to improve clarity. For instance, which release of ISSM did AWI branch from? Was there major development(s) made since then and if so add a reference.
AWI-ISSM denotes the AWI application of ISSM and not any model development. We are working with the developer version of the code, no new branch or anything like that. Differences between different applications of the ISSM's code occur due to several choices,

e.g. ISSM version, initialization technique, choices of boundary conditions, relaxation strategies, grid resolution, employed approximation to Stokes flow, reference SMB, reference year of the initial state etc. (see Tab 3 and Appendix A in Goelzer et al. (2020).) AWI-ISSM has its own choices for all these components. We just wanted to say AWI-ISSM is not equal to any other application of the ISSM model that contributed to ISMIP6. Most likely, differences occur due to model characteristics and not due to the ISSM version used. We add in "code availability" which version of ISSM is used.
However, as we focus now more on the grid-dependency rather than on model description, we dropped to outline differences between the ISSM contributions to ISMIP6.

Line 99: Blatter-Pattyn is a very expensive model to run. Please clarify what you mean by "balancing computational cost", are you referring in comparison to full Stokes?
It is costly compared to SIA and SSA, but cheap compared to Stokes. We rewrote the sentence here and gave a few more details above. See answer to Line77. The sentence here reads: *"Here, we make use of the BP approximation to obtain a most accurate solution."*

Line 102: please add citations for the characteristics of the model (Glen's flow law, temperature dependent rate factor ... ).
Done.

Page 5: Line 104: add "," after "base".
Done

Line 104: please add a citation for this form of sliding law. Also, please clarify your choice of sliding law. This formulation is typically avoided as it can grow unboundedly (schoof 2005). Also, it would be good to provide a map of the k^2 friction coefficient.
Thanks, that's a good point. The choice of the sliding law is based on a long history started a few years ago. We are aware of this limitation and aim to switch to another type of sliding law in the future that considers Iken's bound. We add a map of the obtained friction coefficient k^2 in the supplement (Fig. S3). The section here is rewritten (see lines 100-109 of the new version of the manuscript) by including a reference for the friction law and give a motivation for the choice of this law.

Line 109: "At lateral ... " The sentence is confusing, please reword.
Done.

Line 111: Please provide a citation or link for EPSG:3413 grid.
Done. We dropped "EPSG:3413 grid" as we think this is unnecessary information.

Line 122: Please indicate if the grid is fixed throughout the simulation or evolving.
We added *"... which remains fixed in time".*

Line 124-125: typically, modelers think of high resolution being the smallest mesh size used in a model and the coarse (low) resolution being the biggest one. It is less confusing for RESmin to be the coarsest resolution and RESmax to be the highest.

Yes, that might be true. But we aimed to follow the same conventions given in Goelzer et al. (2018, Tab. 3) and Goelzer et al. (2020, Tab. 4). As a compromise, we changed RESmin to REShigh and RESmax to RESlow.

Line 127: "Additionally, we ... " This information is out of place here and should be omitted.
Done. Sentences are dropped.

Page 6: Line 136: The sentence here contradicts the title of section 3. Maybe rename section 3 as "Forcing experiments" or something similar, and simply state that you are following the ISMIP6 experimental design.
You are right. We renamed it to "Future forcing experiments".

Line 138: I suggest writing "Slater et al. (2019a, b)" similarly to what you did on page 9 line 222.
Done.

Line 139-142: Why is it necessary to mention initMIP here?
We have completely shortened the paragraph here and initMIP does not occur anymore.

Page 7: Line 152: there is also a projection control experiment that starts at the end of the historical run. Have you run it?
Yes, we have run the projection control experiment. See answer to general comment 1.

Line 155: "The ensemble ... " I believe it refers to the ensemble from ISMIP6? If so this sentence does not add any value to the paragraph.
The sentence does not occur any longer. A reference to the ISMIP6 ensemble is only given in the Introduction.

Line 159: Please briefly recall how low, median, and high oceanic forcing were defined.
This is explained later in the text. We added a reference to the section below.

Paragraph 3: "Conducted projection ... " This paragraph is out of place and should be combined somehow with section 3.3.2. The definitions of the runs (which are highlighted in Table2) could be given at the beginning of the result section.
We have restructured the text to better group the information.

Page 8: Line 191: "That means ... " This sentence is confusing. Do you mean that grounded and floating ice cells are not allowed to retreat? If so it restricts the purpose of the historical run. Please clarify.
No, grounded and floating points within the ice extent are allowed to advance and retreat. Here we say that the ice front is fixed. And yes, the historical scenario is restricted in its purpose because we are omitting the response of the outlet glacier due to a changing calving front (which is known as a major driver e.g. for causing a rapid increase of ice discharge (Bondzio et al., 2017)). However, we put not too much effort into reproducing the historical mass loss accurately as this is beyond the scope of this paper. We have rephrased the paragraph (see lines 180-185 in the new version of the manuscript).

In the description of the "Ice flow model ISSM" we mentioned: *"However, at most locations the grounding line coincides with the calving front. Except for the floating tongue glaciers Petermann, Ryder and 79° North, the sub-grid schemes at the grounding line will not apply. The treatment of the calving front evolution depends on the experimental setup and is explained in Sect. 2.3 and 2.4.2."*

Page 9: Line 224: "The imposed ... " Is this sentence supposed to explain how the prescribed calving front retreat was obtained? If so, say so.
We intended to say that the prescribed calving front retreat must be interpreted as a superposition of several mechanisms. We have rewritten the lines to: *"When employing this parameterization the calving front, retreat and advance of marine-terminating outlet glaciers is directly prescribed as a yearly series of ice front positions. (i.e., is not a result of ice velocity at calving front, calving rate and frontal melt that is used to simulate the calving front position)."*

Page 10: Line 228: "This enables ... " See the general comments. Additionally, this statement is ambiguous because you are using an unstructured grid. While you can compare the results from the simulation using different grids, you cannot claim your comparison to be consistent to grid resolution. Please rephrase.
You, are right this claim is misleading. We have rephrased the sentence (see "Comparability of experiments", Lines 232-244 in the new version of the manuscript).

Line 237: The title of section 4.1 reads "Initial state". This title Is confusing. Typically, the initial state is the one obtained at the end of the inversion procedure and the one used as initial condition for the historical and control runs. Please rephrase.
You are right. We renamed the section to *"Historical scenario".*

Section 4.1: this section contains information that should be stated in section 3.1 such as the restriction of the calving front during the inversion procedure ...
Done. We moved the last sentence from section 4.1 to section 3.1.

Page 11: Line 261: "Similar as ... " There is no need to repeat this sentence here since the MSD metric is used again.
We dropped the sentence.

Line 268: "As the ice ... " Please discuss further the reason of keeping the calving front fixed throughout the historical run.
See answer to general comment 6 above.

Line 274: "with the control" The projection runs should be corrected with a projection control run instead which is not discussed in this paper.
See answer to general comment 1 above.

Line 275: "in the absence of additional forcing" This defines the control run. It is an unnecessary repetition.
Done.

Line 276: " ... as a prediction of actual behavior ... " This is out of place because the text is talking about the control and have not induced any forcing yet. Please rephrase.

*Indeed, we are talking about the control and we aim to stress that the response should not be erroneously interpreted as ongoing/observed mass-loss trends. We aim to explain here how the model drift must be interpreted.*

Page 12: Line 279: replace "simulation" with "simulations".
*Done.*

Line 282: replace "with" with "to".
*Done.*

Line 282: "(see above)" Please refer to a section for clarity (unless you are referring to the mass gain numbers?).
*Done.*

Page 14: Line 304: replace "compared the total" with "compared to the total".
*Done.*

Line 307: remove the repetition of "the".
*Done.*

Line 311: replace "RCP8.5-Rnone" with "RCP8.5-Rnone and RCP8.5-Rlow"?
*Done. Rewritten to: "The finer resolutions tend to produce more mass loss in 2100 for the RCP8.5-Rmed/high and OO-Rmed/high experiments. An inverse behaviour is determined for the RP8.5-Rnone experiment. The trend in the RCP8.5-Rlow experiment is not clear."*

Line 316: replace "lesser than" with "less than".
*Done.*

Page 15: Line 347: reword "early in the century an increase" with "an increase early in the century".
*Done.*

Page 16: Line 357: replace "worth to mention" with "worth mentioning".
*Done.*

Line 359: "remains fixed in time ... " See my general comment.
*See answer to general comment.*

Line 366: replace "reduce" with "reduces,".
*Done.*

Line 367: replace "not obvious" with "non obvious".
*Done.*

Line 368: remove "come into play".
Done.

Line 369: "The general picture ... " Please rephrase.
Done. The sentence now reads: *"The responses of most of the outlet glacier reveal the deduced grid-dependent behaviour where higher resolutions cause an enhanced discharge."*

Page 17: Line 378: "To study ... grid size" Please rephrase.
We have the sentence slightly rewritten to: "In order to investigate whether the response behaviour is an effect by purely reducing the grid size, we repeated the OO-Rhigh and RCP8.5-Rhigh experiments with a G1000 simulation using re-gridded bed topography and friction coefficient from the G4000 initial state (simulations are not shown)."

Line 392: replace "together an increase" with "together causing an increase"?
Done.

Line 392: replace "thinning an acceleration" with "thinning and acceleration"?
Done.

Line 392: "The transient ... " Please rephrase the end of this sentence.
Done. Sentence is rewritten to: "*The transient evolution reveals further that thinning and acceleration propagate faster and farther upstream in the finer resolution."*

Page 18: Line 400: replace "nasal" with "basal".
Done.

Line 405: add "we" before "find".
Done.

Line 416: please rephrase end of sentence.
Done.

Line 423: replace "it is worth to investigate this influence isolated" with "it will be worth investigating this influence only"?
Done.

Page 19: Line 428: replace "in numerous cases" with ", in numerous cases,"?
Done

Line 431: replace "assessing the importance of it" with "assessing its importance"?
Done.

Line 434: remove "thus"?
Done.

Tables:

Table 1: Is the computational time listed here for all the experiments or simply for the 86-year run after the historical run?

The table caption states that the computational times are based on a projection run. Following the experiment abbreviations introduced in the overview (Page 7, Line 152), the time span of projection should be clear -> 86-years.

Figures: In the relevant figures, please add a black contour for the grounding line.

In Figs 4 and 6 we add a contour for the grounding line. Depending on the used colormap, the color of the contour in each figure changes. In Fig 10 drawing more lines would be confusing. Also, the grounding line should be identified from geometry alone.

Figure 2: replace "G8000" with "G4000".

Done.

The small ice cap above 79N should not be present for consistency with the text and the other figures in the paper.

Done. The updated figures now present grid resolution only within the initial ice margin.

Figure 5: it should really be Figure 6 since its reference appear after figure 6 in the text.

Thanks, we changed the labels.

Figure 6: it should be relabeled Figure 5 (see figure 5 comment above).

Thanks, we changed the labels.

Figure 9: the subfigure labels b and c are misplaced. What are the units for Year? (I have never seen CE before as a unit).

Subfigure labels are correct. We adopted the figure caption for clarification, given that it caused some confusion.

CE stands for Common Era (https://en.wikipedia.org/wiki/Common_Era) and it is widely used.

Figure 10: the x-axis is labeled "distance". What is it relative to? Please add this reference to the figure.

The distance is a measure of the length along a flow line. We were thinking about to label it as "distance to initial ice front position", but that will not work, because of the different extents of each grid resolution. So, we could set zero distance somewhere, but this would still be an arbitrary point. So we suggest leaving it as is. The distance just helps to identify the correct dimension. We clarified in the caption, that the distance is relative to an arbitrary point.

Also, please try to increase the font size of the labels as they are difficult to read on printed paper.

Done.

[revised manuscript text omitted]

---

## Referee Report (RR1)

**Subject**: Sensitivity of Greenland ice sheet projections to spatial resolution in higher-order simulations: the AWI contribution to ISMIP6-Greenland using ISSM.

I am pleased with this version of the paper and I thank the authors to have followed the guidelines of the reviewers. This paper adds to the comprehension of ice sheet modeling and should get published.

I have added a few minor comments that I caught while reading this version.

**Comments**:

General comment about the figures: When comparing G750 and G4000, it would be nice to be consistent throughout the paper which one is placed on the left-hand side and which one is place on the right-hand side. It makes switching from figures to figures easier.

Line 17 (after abstract): remove "Copyright statement"

Page 5 line127: go back to it after reading sect 2.4 and 2.5.2.

Page 5, last paragraph: this strategy is fine for time scales short enough that will allow the grounding line not to retreat too excessively and remain in the highly resolved part of the grid.

Page 6, figure 2: You forgot to label (a) and (b) in your figures to refer to G750 and G4000 (it was there in the first version of the paper). Also, in all your other figures comparing these 2 resolutions, you clearly label G750 and G4000 inside the figure boxes. I would do the same here for clarity.

Page 7, figure 3: Remove "See" in last sentence of caption.

Page 10, line 220: "When employing…" a word is missing between "parameterization" and "the" (maybe "to" or "at").

Page 11, line238: Replace "On the hand" by "On one hand".

Page 11, line 244: the ";" after "runs" is out of place, you don't need it here. Also, replace "not the case" by "not be the case".

Page 13, line 283: in your case, you have not initialized your model to be at steady state but rather match observations. It happens that the GrIS was in a steady state for some time and matching your observation data set might lead to your model initial state to be in steady state. That said, it should not be surprising if your model experiences some drift during your control experiment (as it does for the coarse resolution).

Page 16, line319: Replace "RP8.5-Rnone" by "RCP8.5-Rnone".

---

## Author Response (AR2)

We would like to thank the reviewers for their constructive comments during the review process that helped to improve the manuscript 'Sensitivity of Greenland ice sheet projections to spatial resolution in higher-order simulations: the AWI contribution to ISMIP6-Greenland using ISSM'. We have revised the manuscript accordingly. Please find below the reviewer's comments in black and a point-by-point response in blue.

**Reviewer 1**

The revised manuscript is interesting and the presentation is coherent and clear. I find that my points have generally been addressed well. I have a few minor comments/corrections:

Abstract, line 5: "to this end", I suggest removing.
Done.

Line 34: The reference to Nowicki et al. 2020b is mentioned before Nowicki et al. 2020a.
Done.

Line 40: The abbreviation "ISM" is not explicitly explained, except as part of "ISMIP6".
Maybe you overlooked it, but ISM is already explained at its first occurrence in Line 32.

Line 95: The abbreviation "BP" is not explained.
You are right. Also, the citation to Blatter 1995 and Pattyn 2003 was get lost in this version. As the abbreviation BP is not further needed in the text we write "Blatter-Pattyn" and add the citations.

Line 238: "On the hand" replace with "On one hand".
Done.

Line 282: "Tab. 3" change to "Table 3".
Thanks, we changed it in the whole manuscript as the TC guidelines states: "Please note that the word "Table" is never abbreviated and should be capitalized when followed by a number (e.g. Table 4)."

Line 450: The grid dependence of the different models must also depend on the flow approximation in the models, and how the different models treat friction along the sides of the fjords. The PISM model used by Aschwanden et al. uses a combination of SIA and SSA solutions, and I am speculating what the effect resolution would have in the result as it may affect the two solutions differently. I suggest that a few sentences on the flow approximation is included in the discussion.
We add a short paragraph on this topic. However, we kept this paragraph very general as we aimed to avoid to introduce the acronyms like SIA and SSA; this would require further explanation of the underlying approximations.

Line 484: Please add the paper Rathmann et al. 2017 to the list of references, and add a comment that surface melt can both affect the basal conditions and the frontal melt with different effects on the flow, as investigated by Rathmann et al. investigated for the response of the Zachariae and 79N outlet glaciers to seasonal surface melt.

Rathmann, N. M., C. S. Hvidberg, A. M. Solgaard, A. Grinsted, G. H. Gudmundsson, P. L. Langen, K. P. Nielsen, and A. Kusk (2017), Highly temporally resolved response to seasonal surface melt of the Zachariae and 79N outlet glaciers in northeast Greenland, Geophys. Res. Lett., 44, 9805–9814, doi:10.1002/2017GL074368.

We add the citation to the list and add „A feedback …  and calving by filling up crevasses."

**Reviewer 2**

**Subject**: Sensitivity of Greenland ice sheet projections to spatial resolution in higher-order simulations: the AWI contribution to ISMIP6-Greenland using ISSM.

I am pleased with this version of the paper and I thank the authors to have followed the guidelines of the reviewers. This paper adds to the comprehension of ice sheet modeling and should get published.

I have added a few minor comments that I caught while reading this version.

**Comments**:

General comment about the figures: When comparing G750 and G4000, it would be nice to be consistent throughout the paper which one is placed on the left-hand side and which one is place on the right-hand side. It makes switching from figures to figures easier.
Done. G4000 is now consistently located on the left-hand side while G750 on the right-hand side.

Line 17 (after abstract): remove "Copyright statement"
Done.

Page 5, last paragraph: this strategy is fine for time scales short enough that will allow the grounding line not to retreat too excessively and remain in the highly resolved part of the grid.
You are absolutely right. Of course, a remeshing is needed once the grounding line retreats out of the highly resolved part. We add an explanatory sentence.

Page 6, figure 2: You forgot to label (a) and (b) in your figures to refer to G750 and G4000 (it was there in the first version of the paper). Also, in all your other figures comparing these 2 resolutions, you clearly label G750 and G4000 inside the figure boxes. I would do the same here for clarity.
Done.

Page 7, figure 3: Remove "See" in last sentence of caption.
Done.

Page 10, line 220: "When employing..." a word is missing between "parameterization" and "the" (maybe "to" or "at").
Right. We added "to".

Page 11, line238: Replace "On the hand" by "On one hand".
Done.

Page 11, line 244: the ";" after "runs" is out of place, you don't need it here. Also, replace "not the case" by "not be the case".
Done.

Page 13, line 283: in your case, you have not initialized your model to be at steady state but rather match observations. It happens that the GrIS was in a steady state for some time and matching your observation data set might lead to your model initial state to be in steady state. That said, it should not be surprising if your model experiences some drift during your control experiment (as it does for the coarse resolution).
We agree and have rewritten the sentence to: "As we have not initialized our model to be at steady state the transient response in the ctrl experiment (thin coloured lines in Fig. 6 should not be interpreted as a prediction of actual future behaviour, the ctrl run rather confirms that each model has achieved a high degree of equilibration, which is reflected with a low rate of volume change."

Page 16, line319: Replace "RP8.5-Rnone" by "RCP8.5-Rnone".
Done.

[revised manuscript text omitted]

---

## Author Response (AR3)

Dear Robin,

Thank you for your comments. We made the changes accordingly but do not follow your optional comment (see answer below).

Please note, that we also made some minor changes that were not requested:

(1) The Goelzer et al. (2019, in review) is now Goelzer et al. (2020b), because it changed from 'in discussion' to 'published'. Therefore Goelzer et al. (2020) is now Goelzer et al. (2020a).

(2) We clarified the *code-and-data-availability* section. We changed the intended data archive from Pangaea to zenodo as Pangaea is currently slower in publishing data due to COVID-19 constraints. Please note, the zenodo DOI is reserved but the data will be published soon.

(3) Heiko Goelzer received a third affiliation; he recently moved to Norway.

Please find our responses in blue below.

Kind regards,
Martin and co-authors
* * *
Comments to the Author:
Figs 2 and 7: panels are labelled a=G4000, b=G750, caption says they are the other way round.
Thanks. We changed it accordingly.
It would be worth checking the labelling of panels in figs 3, 12 and 13 too, although their captions do not say which should be which and they look right to me.
The labelling in Figs 3, 12 and 13 are correct.

line 9,11,12: in an abstract, for clarity and reproducability in whatever system may be indexing it, I would write out "approximately" instead of using "~", and "below" instead of "≤". The typesetter may disagree.
We changed as suggested.

line 7: it would be clearer if this had something up front to clearly explain that you're talking about a climate forcing for those who might be unfamiliar with climate model names, eg "We run the simulation based on the ISMIP6 core climate forcing from the MIROC5 AOGCM [...]" (although neither of the acronyms GCM nor AOGCM have been explained in this abstract, and should be if you're going to use them).
Done.

----------optional

Reviewer 1 commented
Line 450: The grid dependence of the different models must also depend on the flow approximation in the models [...] I suggest that a few sentences on the flow approximation is included in the discussion.

and you replied
We add a short paragraph on this topic [...] However, we kept this paragraph very general

This addition isn't really a paragraph if we're honest, and I for one would find it really useful if you documented some ideas about how different approximations might behave in different ways as their grid resolutions change. After all, sensitivity of the modelling to grid-resolution is the primary topic of your study and if you can help more people see how your results might relate to their modelling setups then that's good for everyone. I'll leave it up to you to decide whether to expand any further here.

You are right; we just added a sentence instead of a paragraph. In general, we agree that it would be interesting and important to add more on this topic. Despite the different grid resolution and flow approximations, the other studies rely on different scenarios, numerics etc, which makes a firmer comparison to other studies very speculative. Therefore, we would like to leave it as is.

[revised manuscript text omitted]